environmental science/palaeontology/ecology

palaeoecology, plant–insect interactions, fossil leaf, endemism

**Author for correspondence:**
Benjamin Adroit
e-mail: benjamin.adroit@gmail.com

# A case of long-term herbivory: specialized feeding trace on *Parrotia* (Hamamelidaceae) plant species

Benjamin Adroit[1], Xin Zhuang[2], Torsten Wappler[3], Jean-Frederic Terral[4] and Bo Wang[1]

[1]State Key Laboratory of Palaeobiology and Stratigraphy, Nanjing Institute of Geology and Palaeontology, Chinese Academy of Sciences, 39 East Beijing Road, Nanjing 210008, People's Republic of China
[2]College of Life Sciences, Nanjing University, 22 Hankou Road, Nanjing 210093, People's Republic of China
[3]Hessisches Landesmuseum Darmstadt, Darmstadt, Germany
[4]Institut des Sciences de l'Evolution, UMR5554 Université de Montpellier, CNRS, IRD, EPHE, Place Eugène Bataillon, 34095 Montpellier cedex 05, France

BA, 0000-0003-3693-9581

Interactions between plants and insects evolved during millions of years of coevolution and maintain the trophic balance of terrestrial ecosystems. Documenting insect damage types (DT) on fossil leaves is essential for understanding the evolution of plant–insect interactions and for understanding the effects of major environmental changes on ecosystem structure. However, research focusing on palaeoherbivory is still sparse and only a tiny fraction of fossil leaf collections have been analysed. This study documents a type of insect damage found exclusively on the leaves of *Parrotia* species (Hamamelidaceae). This DT was identified on *Parrotia* leaves from Willershausen (Germany, Pliocene) and from Shanwang (China, Miocene) and on their respective endemic modern relatives: *Parrotia perisca* in the Hyrcanian forests (Iran) and *Parrotia subaequalis* in the Yixing forest (China). Our study demonstrates that this insect DT persisted over at least 15 Myr spanning eastern Asia to western Europe. Against expectations, more examples of this type of herbivory were identified on the fossil leaves than on the modern examples. This mismatch may suggest a decline of this specialized plant–insect interaction owing to the contraction of *Parrotia* populations in Eurasia during the late Cenozoic. However, the continuous presence of this DT demonstrates a robust and long-term plant–herbivore association, and provides new evidence for a shared biogeographic history of the two host plants.

# 1. Introduction

An ecosystem is a set of abiotic environmental conditions with communities of organisms living and interacting therein. Interactions between organisms contribute to a fragile equilibrium within ecosystems [1–3], and this balance can be drastically disturbed by modern human activities [4–6]. Global environmental change is expected to affect plant–insect associations in various ways, as insects play significant roles such as herbivores [7] and pollinators for crop production [8]. These interactions are the result of millions of years of evolution [9–11]. Plant–herbivore interactions are of particular importance for terrestrial food webs that sustain biodiversity and ecosystem balance [12–14]. Variation in the style and quantity of herbivory depends mostly on abiotic parameters, primarily climatic conditions [9,15–17]. Consequently, it is not surprising that many studies have measured significant changes in the patterns of herbivory on fossil leaves through geological time as environmental conditions have changed [16,18–25].

Recent research has been carried out on some late Cenozoic floras from Europe using standardized damage type (DT) nomenclature. This research includes work on the famous Lagerstätte of Willershausen in Germany [26]. During the identification of herbivory traces on fossil leaves of Willerhsausen (i.e. 8073 fossil specimens analysed), an insect DT was recorded exclusively on fossil leaves similar to *Parrotia persica* (DC.) C. A. Mey. These leaves were the most abundant within the fossil assemblage [26,27] and hosted many examples of this specific insect feeding trace [26]. Parallel to this study, another investigation was conducted mostly on modern leaves of *P. persica* from the Hyrcanian forest region [28]. During this study, herbivory traces similar to those on the fossils were observed on living *P. persica* at Aliabad-e Katul, Pasand and Molla Kala in Iran (figure 1). Lastly, within the framework of another project on fossil plants from the Lagerstätte of Shanwang (Miocene), north-east China [31,32], around 1300 leaves were studied and the same insect feeding trace was identified on just one *Parrotia* leaf.

This study describes the type of the plant–herbivore interaction found exclusively on leaves of the two known *Parrotia* species. In addition to the fossil and modern materials mentioned previously, some modern leaves of *Parrotia subaequalis* were also measured in the Yixing forest (China), where one of the small endemic populations of *P. subaequalis* still exists [33]. We discuss how such external leaf feeding remained unchanged for 15 Myr across Eurasia, in the face of major environmental changes. Finally, we discuss how this discovery provides new perspectives on the evolution of plant–insect interactions.

# 2. Material and methods

## 2.1. Fossil record of *Parrotia*

*Parrotia* was present in East Asia and possibly North America during the Eocene and seemed to spread to western Eurasia across Central Asia during the Oligocene [34]. During the early Oligocene the genus was present in Kazakhstan from where it disappeared during the Miocene [35]. *Parrotia pristina* (Ettingshausen) Stur and *Parrotia fagifolia* (Göppert) Heer were described from Europe. These names were also used for Palaeogene and Neogene leaf fossils of East Asia [36,37]. In addition, *Fothergilla* Hu & Chaney [38] was described from the early to middle Miocene Shanwang flora of China. Based on the morphological similarity with the extant *Shaniodendron subaequalis* (=*Parrotia subaequalis*), *Fothergilla virburnifolia* was later transferred to *Shaniodendron viburnifolium* (Hu and Chaney) Wang & Li [39]. In Europe and Kazakhstan, fossils assigned to *Parrotia* are commonly called *P. pristina* (*P. fagifolia* being a junior synonym). In East Asia, the nomenclature is somewhat unclear: *S. viburnifolium* should be treated as *Parrotia virburnifolia* based on the current taxonomic treatment of *Shaniodendron* as a synonym of *Parrotia* [40]. At the same time, this name competes with the earlier name *P. pristina*, which also has been used for East Asian fossils. For practical reasons, in this study we refer to the European fossils as *P. pristina* and to the East Asian examples, as *P. viburnifolia*.

## 2.2. *Parrotia persica* (DC.) C. A. Mey

*Parrotia perisca* is a deciduous tree 8–25 m tall [41]. The leaves are oblong to obovate, up to 15 cm long and 6 cm wide, with 5–8 pairs of secondary veins [27,41,42]. Nowadays, *P. persica* exists only in the Hyrcanian forest south of the Caspian Sea (Iran, Azerbaijan). The leaf shape of *P. persica* is very consistent despite there being size variability in leaves throughout the Hyrcanian forest owing to variable local abiotic

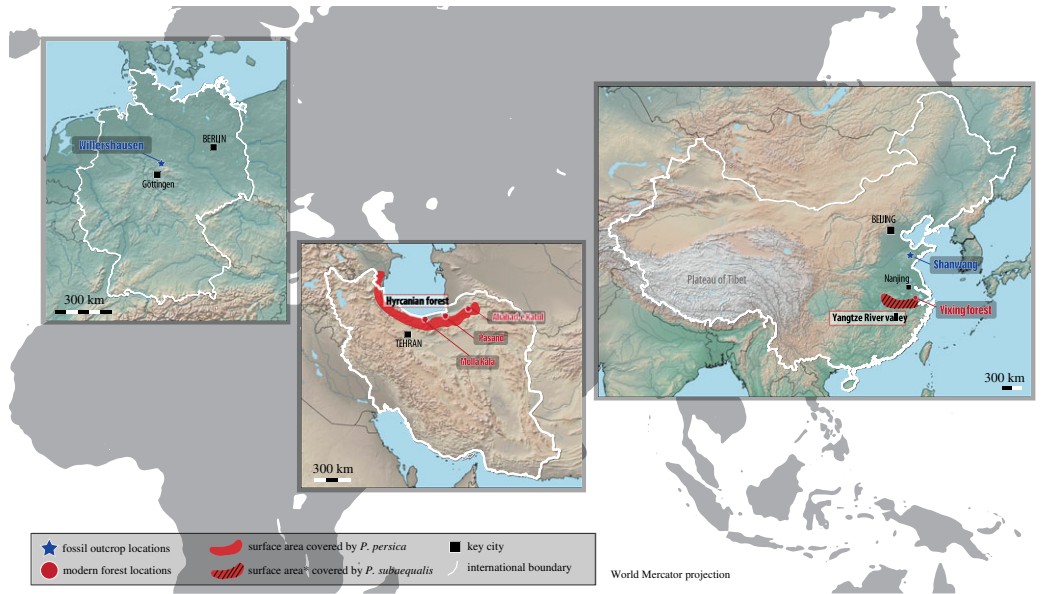

**Figure 1.** Eurasian locations including both fossil and modern occurrences of *Parrotia* that contain examples of the damage type DT297. The distribution for *P. subaequalis* in China has been drawn in one large area for clarity. In reality, the distribution of *P. subaequalis* in the Yangtze River valley is very fragmented, only small isolated populations occur in the valley. More details are provided by Geng *et al.* [29] and Li & Zhang [30].

conditions [43]. *Parrotia persica* is an Arcto-Tertiary relict species [34]. Its close fossil relatives were very common in European forests during the Neogene [42,44–50]. In this study, fossil leaves from Willershausen (Pliocene) in Germany and modern leaves from the Hyrcanian forest in northern Iran were studied comparatively (figure 1).

### 2.2.1. Willershausen, 3 Ma, Germany

Willershausen is a Lagerstätte in the centre of Germany, close to Göttingen (figure 1). It is a lacustrine clay pit containing more than 130 fossil plant species including many leaves of *Zelkova zelkovifolia*, *Carpinus orientalis* Mill., *Carya minor* Schenk and *P. pristina* [27,48,51] the last of these is the most abundant in the fossil plant assemblage of Willershausen [26]. The palaeoforest represented by this fossil leaf assemblage is dated around 3 Ma; MN 16/17 [34,52]. Adroit *et al.* [26], analysed 8073 leaf specimens of which 517 were attributed to a fossil relative of *Parrotia*.

### 2.2.2. Hyrcanian forest, modern, northern Iran

In terms of plant species richness, the Hyrcanian forest region (figure 1) is considered a good modern analogue of the European forests of the late Cenozoic [42,53], such as that represented by the Willershausen Lagerstätte. The Hyrcanian forest region is a refuge for several extant Arcto-Tertiary plant species that are endemic to this area (such as *P. persica*) [34,42,54,55]. This forest extends from Golestan National Park (northeastern Iran) to eastern Azerbaijan, and is bordered by the Caspian Sea to the north and the Alborz mountains to the south, encompassing 1.85 million ha [56]. Adroit *et al.* [28] collected and analysed 2160 leaves of *P. persica* and observed additional leaves from other species (such as *Zelkova carpinifolia* (Pall.) K. Koch, *Quercus castaneifolia* C. A. Mey, *Acer cappadocicum* Gled.), which commonly co-occur with *Parrotia*.

## 2.3. *Parrotia subaequalis* (H. T. Chang) R. M. Hao and H. T .Wei

Similar to its sibling species [57,58], *P. subaequalis* is a large shrub or small tree, 5–10 m tall [57]. Rarely, it reaches up to 20 m tall with pruning and staking, as evident in a village on Qingliang Peak, Linan, China. Leaf blades of *P. subaequalis* are mostly broad-obovate or elliptic, 4–6.5 cm long, and 2–4.5 cm wide, and thinly leathery [59]. *Parrotia subaequalis* is a Cenozoic relic plant species endemic to eastern China [29].

Fossil specimens from Miocene strata indicate the former distribution of *Parrotia* in Shanwang, Shandong Province, northeastern Central China [60,61] and in Huadian, Jilin Province, northeastern China [62]. Its population size severely decreased during Quaternary glaciations [63]; the modern species has a narrow and scattered distribution on Mt Qinling-Dabie and Mt Tianmu (China). *Parrotia subaequalis* was described from Yixing, Jiangsu province as *Hamamelis subaequalis* H. T. Chang and later transferred to the monotypic genus *Shaniodendron* [64]. Subsequently, flower morphology [60] and a molecular phylogenetic study [58] suggested that *Shaniodendron* should be included within *Parrotia* resulting in the name *P. subaequalis*. The modern leaves of *P. subaequalis* came from the Yixing forest in eastern China (figure 1).

### 2.3.1. Shanwang, 18–15 Ma, China

Shanwang is a Lagerstätte [31] containing a diverse assemblage of organisms dominated by angiosperms [31,32]. It is located in northeastern China, in Shandong province (figure 1).

According to various dating methods, the Shanwang deposit is early—middle Miocene [65], i.e. 18–15 Ma [31,66–69]. Both pollen and fossil leaf studies indicate the presence of *Quercus*, *Pterocarya*, *Ulmus*, *Populus*, *Fraxinus*, *Carpinus* and *Betula* [70]. They represent an evergreen broad-leaved and mixed deciduous forest [70]. The fossil collection from this deposit is stored in the Nanjing Institute of Geology and Palaeontology (Nanjing, China) and includes 1298 leaves, of which 40 are attributed to *Parrotia*.

### 2.3.2. Yixing forest, modern, eastern China

The modern Yangtze River valley is an appropriate environmental analogue of the Shanwang Miocene site, although the Shanwang palaeoforest may have experienced lower annual temperatures including possibly colder summers and lower seasonality in rainfall. Yixing Forest Farm, located in the Yangtze River valley (figure 1), is one of the most significant state-owned forest farms in southwestern Jiangsu Province, covering 34 km², with 97% forest coverage. This farm is set in the region of Mt Yili, which is geographically a low-altitude hilly terrain forming the eastern extension of Mt Qinlin-Dabie [71]. There is a small population of *P. subaequalis* trees in the central part of Yixing Forest Farm, with three eminent old trees and around 20 mature individuals. Other small populations occur within and around the farm [29]. In the field, 41 leaves of *P. subaequalis* were sampled.

## 2.4. Observations

All the specimens were studied with a stereomicroscope (Leica EZ4) and a transmitted light microscope (Zeiss AXIO Zoom V.16). They were photographed with a Lumix GX8 mounted on a copy stand. The fossil leaves from the various collections were sampled many years ago and described in previous works. No additional sampling was attempted, as the fossil collections are large enough and because, nowadays, collecting in Willershausen (Germany) and Shanwang (China) is forbidden. All the modern leaves were sampled from the ground (litter) in the Hyrcanian forest region (Iran) and Yixing forest (China). Leaves from the litter are more representative for plant–insect interactions as herbivory is not homogeneously distributed throughout the tree and the whole spectrum of leaf DTs is best captured when leaves from all parts of the tree including the canopy are considered [72–75]. Moreover, to collect a fallen leaf from the litter means collecting after the whole lifespan of this leaf, and then no more herbivory can happen. Lastly, the leaves from the litter represent at least a part of the taphonomy process. For those reasons, the leaf litter is better for the standardization of samples for the whole study.

## 2.5. Terminology

Currently, the main reference to identify and classify the plant–insect interactions in the fossil record is the '*Guide to Insect (and Other) Damage Types on Compressed Plant Fossils*' [76]. This guide subdivides herbivory traces on leaves into seven functional feeding groups (FFGs): hole feeding, margin feeding, skeletonization, surface feeding, mining, piercing and sucking and galling. For each FFG, numerous DTs are recognized. For each of these DTs, a host specificity index (HS) is assigned that distinguishes between generalist and specialist damage [77]. The determination of this HS index is based on diverse parameters, such as its geographical distribution, plant species diversity affected by the damage, damage quantity, shape variations, among others factors (more details in [25,70]).

## 2.6. Deposition of fossil specimens

Fossil *P. persica* leaves from Willershausen (Germany) analysed in this study are all deposited at the Geoscience Centre of the University of Göttingen (GZG.W collection). The fossil leaves of *P. viburnifolia* (labelled as *P. subaequalis* in the collection) belong to the Nanjing Institute of Geology and Palaeontology, Chinese Academy of Sciences (China).

## 2.7. Measurements

With the help of photography and the software IMAGEJ, each leaf was measured following several parameters, such as length, width and surface area of the leaf blade. Where possible, the width of the petiole was also measured in order to determine the leaf mass per area (LMA) for each specimen. LMA is an index that corresponds to the relationship between the thickness and the density of the leaf [78,79]. Thereafter, the specific damage on each *Parrotia* leaves was recorded and described, and the measurements of the surface area of the damage, the length and the width at three different positions along the damage, and the number of holes, were compared between the two *Parrotia* species and both fossil and modern leaves. Basic statistical tests based on the averages of measurements (Shapiro, Fisher and Wilcoxon) were made in addition to these morphological comparisons.

# 3. Results

At Willershausen, 32 leaves of *P. pristina* had the DT DT297 and we counted 143 occurrences in total. In general, one leaf can include more than one damage occurrence (figure 2). In Shanwang, four leaves of the species *P. viburnifolia* (= *P. subaequalis* fossil relative) had 13 occurrences of this specific DT. In all of the fossil collections, no leaves from other species had DT297. In the modern Hyrcanian forest, despite the large amount of *P. persica* leaves collected, only six had this DT, and in a low quantity as only seven occurrences in total were recognized. In the Yixing forest, damage abundance is even lower; only four occurrences on two leaves of *P. subaequalis* were identified.

Overall, 167 damage occurrences were observed in this study among 43 leaves of *Parrotia* species (figure 2). A large majority of them were observed on *P. pristina* from Willershausen (85%) and then 7% on *P. viburnifolia* from Shanwang. The modern samples of *Parrotia* spp. (i.e. *P. persica* and *P. subaequalis* together) represent 8% of our observations.

## 3.1. Morphological description

The average of the damage measurements has been calculated by counting all the occurrences from all the leaves, but it must be noted that fossil *Parrotia* leaves from Willershausen are the most representative in terms of the size variability for this specific damage (figure 3). This is certainly a consequence of the large quantity of specimens analysed. Nevertheless, we split and compared the average of damage sizes per locality (then per *Parrotia* species) (figure 3).

The damage trace is a curved skeletonization subdivided in a row of several holes. This long, curved chain of small holes usually is less than 1 cm long. However, some of the specimens can reach 1.5 cm in length, but this is quite rare. Individual holes are commonly rectangular with rounded corners. The length of each hole never exceeds more than 1 mm and the width of each hole, i.e. the width of the damage, is around 0.6 mm. There is no variation of the width along the course of the damage. On average, the surface area of the damage is 4.1 mm$^2$ (±1.4) with a length of 5.4 mm (±1.3) and a global width of 0.6 mm (±0.08). The number of holes can vary from 3 to 12, but in most cases it is 5–8.

The small lines that separate individual holes from each other are very thin, commonly inconspicuous or missing. Although these lines are indistinct, it is possible to infer their existence by carefully observing the internal borders of the damage. In some cases, the small lines are missing along longer portions of the damage (figure 2, C2). The margin of the damage is marked by black edges. This black scar is a typical reaction from the leaf after being attacked by insect feeding and makes it possible to distinguish a herbivory trace made by an insect during the leaf's lifespan from a detritivorous trace made after abscission [76,77]. Overall, the path of the damage is not affected by the leaf venation. However, we noted that the damage usually follows a secondary vein instead of removing it. Exceptionally, we

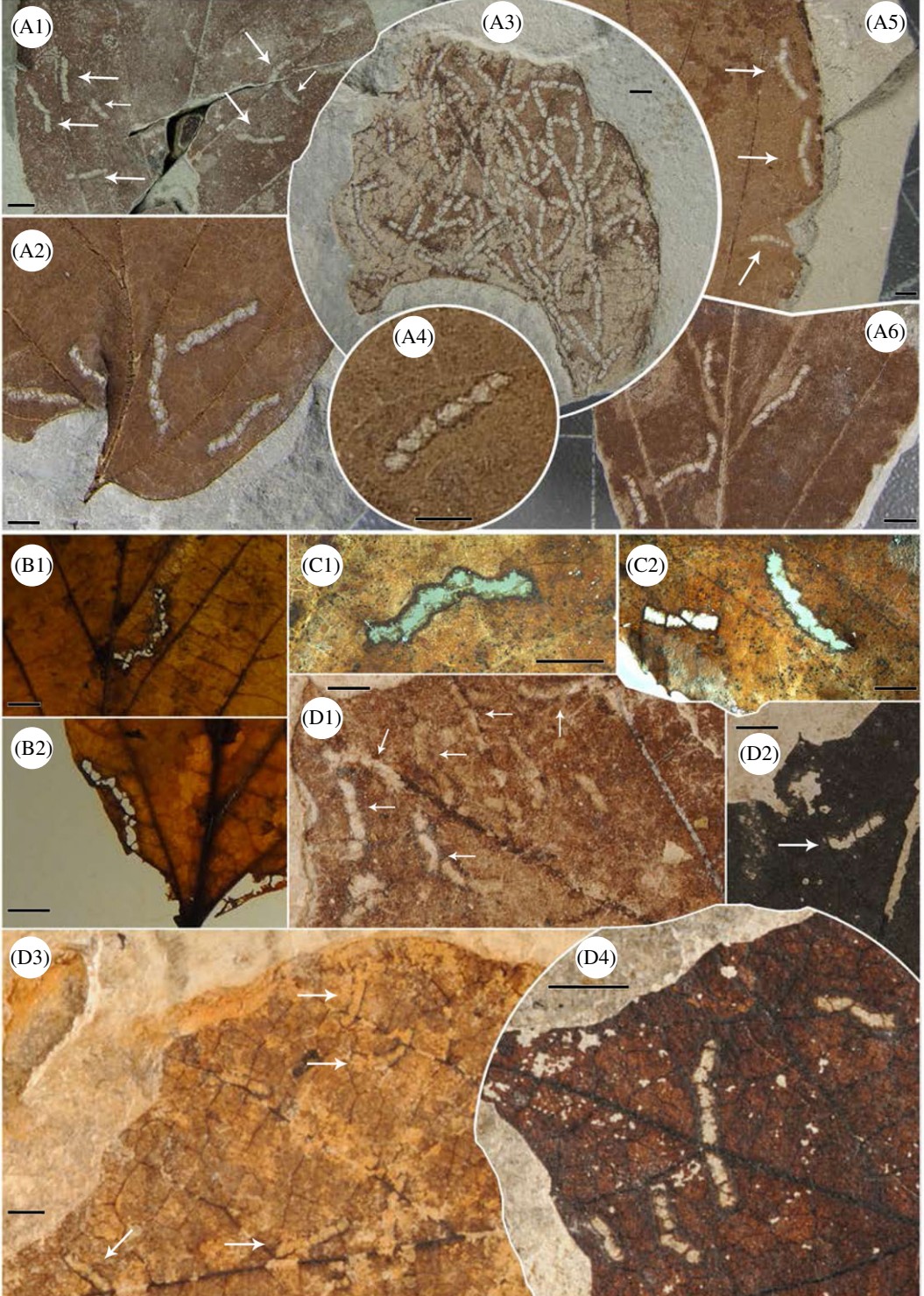

**Figure 2.** Damage type DT297 on every type of leaf attributed to *Parrotia*. A1–A6 Fossil specimens of *P. pristina* from the Pliocene of Willershausen, Germany. The material is deposited in the Geoscience Centre, University of Göttingen (GZG), Germany. B1–B2 Modern specimens of *P. persica* from the Hyrcanian forest (northern Iran), more details are provided by Adroit *et al.* [28]. C1–C2 Modern leaves of *P. subaequalis* from the Yixing forest, Yangtze River area, eastern China. D1–D4 Fossil specimens of *S. subaequalis* (= synonym of *P. subaequalis*) from the mid-Miocene of Shanwang, China. The material is deposited in the collection belonging to the Nanjing Institute of Geology and Palaeontology (NIGPAS), China. Black bars represent 2.5 mm.

noted that damage crosses over the primary vein (figure 2, D4) but without removing the vein. There is usually more than one damage example per leaf blade; a single example of damage per leaf is rare. We also observed some cases in which the entire leaf blade was covered by this damage (figure 2, A3).

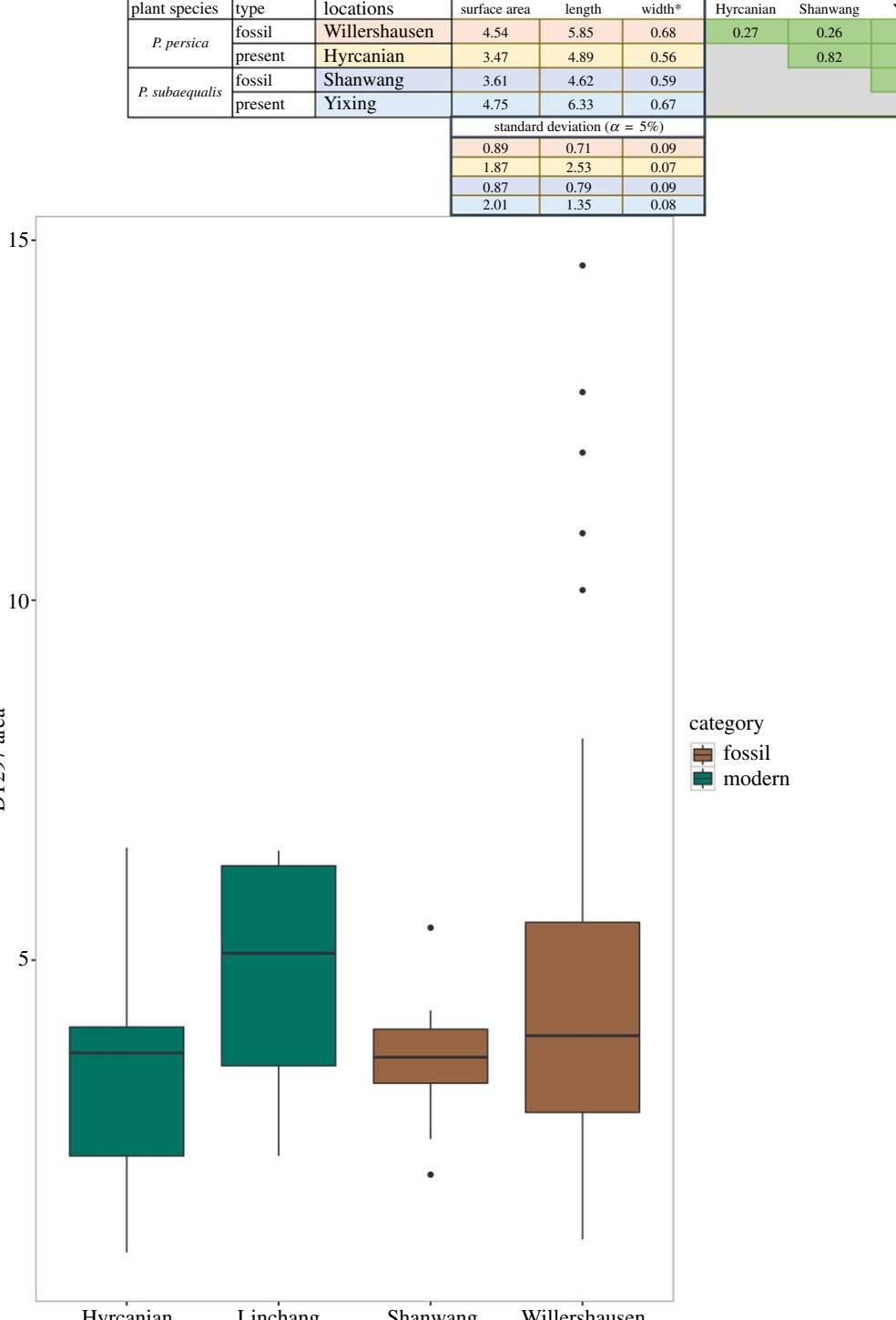

| plant species | type | locations | DT297 measurements | | | Wilcoxon test** (*p*.value) | | |
|---|---|---|---|---|---|---|---|---|
| | | | surface area | length | width* | Hyrcanian | Shanwang | Yixing |
| *P. persica* | fossil | Willershausen | 4.54 | 5.85 | 0.68 | 0.27 | 0.26 | 0.59 |
| | present | Hyrcanian | 3.47 | 4.89 | 0.56 | | 0.82 | 0.53 |
| *P. subaequalis* | fossil | Shanwang | 3.61 | 4.62 | 0.59 | | | 0.35 |
| | present | Yixing | 4.75 | 6.33 | 0.67 | | | |
| | | | standard deviation ($\alpha$ = 5%) | | | | | |
| | | | 0.89 | 0.71 | 0.09 | | | |
| | | | 1.87 | 2.53 | 0.07 | | | |
| | | | 0.87 | 0.79 | 0.09 | | | |
| | | | 2.01 | 1.35 | 0.08 | | | |

**Figure 3.** Box-plots based on the comparison of the average area of DT297 per locality. Green boxes represent modern leaves, brown boxes denote fossil leaves. The upper table provides all measurements made on DT297. A Wilcoxon test comparison has been made on the measurements and the results concerning surface area comparison between each location are presented on the right. Surface area has been chosen as it directly includes the length and width. There is no significant difference between values from these sites ($\alpha$ = 1%).

## 3.2. Host plant

Based on the fossil record, there is little doubt that DT297 is exclusively found on *Parrotia* species. Investigations of fossil leaves from Willershausen were based on around 8000 fossil specimens representing more than 130 plant species/morphotypes. The Shanwang collection consists of around 1300 leaf specimens and includes more than 100 morphotypes.

## 3.3. The specimen reference of DT297

This DT was originally described as a trace fossil *Phagophytichnus catellarius* ichnosp. nov. by Straus [51]. The fossil specimen of *P. pristina* from which *Ph. catellarius* was described belongs to the Willershausen fossil collection from Göttingen. We photographed in high resolution the sample used for the holotype of this DT (electronic material supplementary, S1). The holotype is labelled GZG.W no. 10626 and is located in the Willershausen plant macrofossil collection at Göttingen University, Germany.

## 3.4. Classification

The classification of this DT follows the rules and terminology of the *Guide to Insect (and Other) Damage Types on Compressed Plant Fossils* [76]. This damage is now designated: DT297.

# 4. Discussion

First and foremost, it is important to mention that there is no difference in terms of leaf thickness between the two extant *Parrotia* species or between fossil and modern leaves, as estimated by LMA (electronic material supplementary, S2) based on the method from Royer *et al.* [79]. LMA can be correlated with climatic factors [80–82], leaf nutrient availability [83,84] and, furthermore, can affect herbivory patterns observable on the leaf blade [78,85].

## 4.1. DT297: a new classification for this specialist insect damage trace

Our observations demonstrate that modern *P. persica* and *P. subaequalis* bear the same insect feeding trace (DT297). Morphological descriptions are consistent and statistical assessments support this observation. Indeed, measurements of the surface area, length and the width of this DT between the *Parrotia* species and between fossil and modern specimens do not reveal any significant differences ($\alpha = 0.1$) (figure 3). The statistics may be quite weak owing to the small number of measurements on modern leaves. However, as the statistics did not demonstrate any significant variations of measurements, our results indicate little size variation of the damage.

The DT DT297 (figure 4) is new for the '*Guide to Insect (and Other) Damage Types on Compressed Plant Fossils*' [76] and will be will be considered for the next version of that guide and classified into the 'skeletonization' FFG.

This herbivory trace is exclusively found on *Parrotia* for at least 15 Myr. Indeed, it is important to note that the outcrops mentioned in this study are not the only ones which have been investigated for the present study. Several fossil localities in Eurasia of Cenozoic age have also been investigated but lacked any trace of this DT [20,23,28,75,86–89], either on *Parrotia* leaves or on any other plant species. Such assemblages include the fossil leaves of *Parrotia* from Berga and Bernasso documented by Adroit *et al.* [26]. Further, the method of identification of DTs in the leaf fossil record [76] has now been used for more than 10 years in numerous studies throughout the world and through all geological time periods [18,21,24,25,90–92], and none mentioned feeding traces equivalent to DT297. Hence, DT297 can be considered a highly specialized term of skeletonization with a host specificity index of 3 (HS = 3).

## 4.2. Specialist herbivory pattern for 15 Myr in Eurasia

DT297 provides direct evidence of the continuous relationship between a plant and a herbivore. So far, this is the most ancient herbivory trace specifically identified and still distinctive in the modern flora on the same plant genus. This specific damage has never changed in terms of plant host association or morphological characteristics (shape, size). It has been distinguished from western Europe to eastern Asia over at least 15 Myr; a long period of time and a large geographical area characterized by marked environmental differences.

The warm climate during the Middle Miocene Climatic Optimum (17–15 Ma) [93], followed by progressive cooling during the Middle Miocene Climate Transition (15–13 Ma) [94], and the onset of glacial–interglacial cycles from the Middle Pleistocene Transition (1.2–0.7 Ma) onwards [95] occurred between the first known traces of DT297 and the present. In addition, orogenesis was extremely important in Eurasia, especially with the rise of the Tibet-Qinghai Plateau [96–98], which formed a barrier between eastern Asia (*P. subaequalis*) and the Caucasus—Europe (*P. persica*) during this interval.

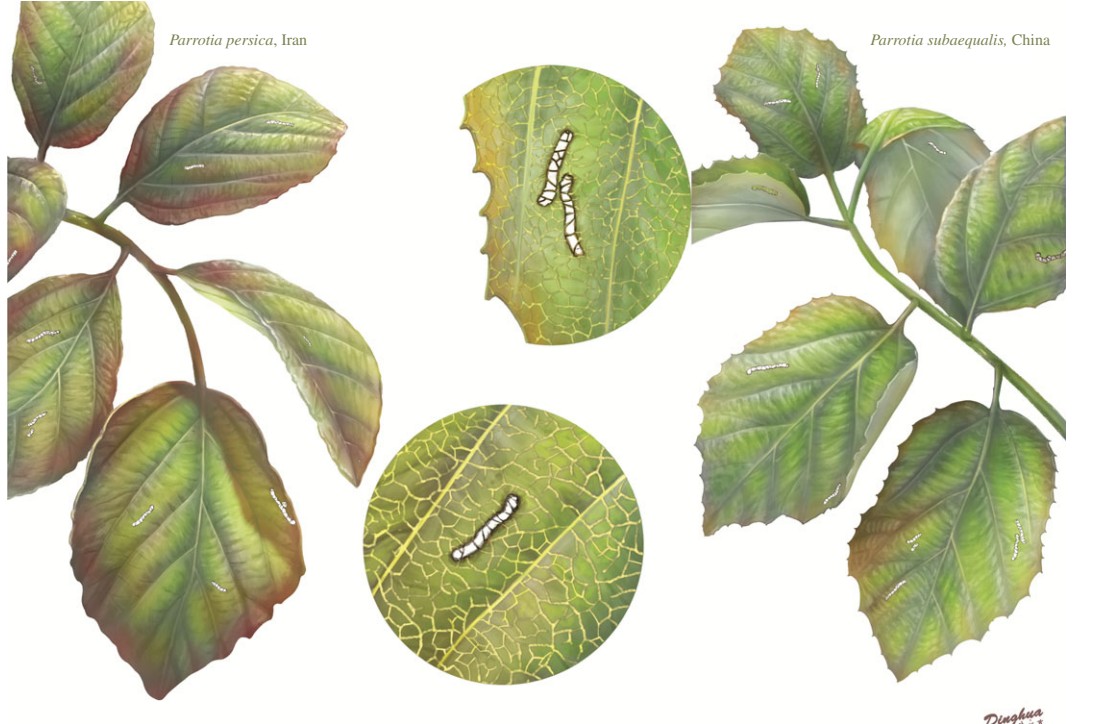

**Figure 4.** Artistic representation of both *Parrotia persica* (left) and *Parrotia subaequalis* (right) bearing the damage type DT297. The illustration was produced by Mr Dinghua Yang from the Nanjing Institute of Geology and Palaeontology, Nanjing, China.

DT297 represents remarkable stasis in a feeding strategy. Numerous studies have demonstrated shifts in herbivory during various geological events [16,23,99,100]. Those herbivory changes are mostly caused by climate variations impacting insect physiology [101–105] and, in some cases, by interruptions of gene exchange between plant and insect species [106,107] created by the emergence of new landforms.

Although the specific damage on *Parrotia* over (at least) 15 Myr can be used to reconstruct the trophic relationships of *Parrotia* in its environment, it is very difficult to determine the insect causing this damage. Straus [51] suggested that this trace fossil could have been produced by Chrysomelidae larvae. Based on our comparisons with known insect feeding from the literature we suggest that DT297 could have been caused by insects belonging to subfamily Galerucinae/Alticinae, probably by *Altica* which was widely distributed in Eurasia from at least the Eocene [108–110]. Both subfamilies are recorded in the Yangtze River valley and the Hyrcanian forest region with some endemic species of these regions [111–116].

The present specific damage shared between *P. subaequalis* and *P. persica* which are today completely isolated from each other, provides direct evidence that they occupied a common ecological niche, which is today separated by vicariance into two geographical areas [117]. The congenerity of the *Parrotia* species between eastern Asia and Caucasus is still not clear based on previous studies and has led some to assign the Asian taxon to *S. subaequalis* (= *P. subaequalis*) for some researchers [39,64]. However, Li *et al.* [118] used internal transcribed spacers from the nuclear gene of Hamamelidaceae and found support for *Parrotia* and *Shaniodendron* as a monophyletic group [118,119]. Additional studies based on the chloroplast gene 'matK' segregated *Parrotia* and *Shaniodendron* as distinct taxa [120,121]. A recent study even describes the whole chloroplast genome of *P. subaequalis* [122]. However, chloroplast markers usually do not reconstruct taxonomic but biogeographic relationships [123–127]. Thus DT297 can contribute to better understanding of the shared biogeographic history of the two host lineages in western Eurasia and eastern Asia and supports the accordance of lineages between these *Parrotia* species presented in this study.

In a continuous coevolution between insect attack and plant defence strategies [128,129], it is difficult to explain why such a specific herbivory mode has never changed over 15 Myr. One hypothesis is that this insect damage can be mutualistic in some cases. Although this is debated within the scientific community, Agrawal [130] demonstrated, based on plant fitness, that certain types of insect feeding could represent mutualistic interactions between the insect and the plant. Moreover, a recent meta-analysis of

hundreds of scientific publications [131] about the 'overcompensation' for insect herbivory also supports this hypothesis. However, to focus only on plant fitness is insufficient to discuss mutualism as a whole, because mutualism also implies an evolutionary history of the plant–animal interaction [132], in which a specific feeding trace, such as DT297, can be considered as a direct evidence.

## 4.3. DT297 more common in the fossil record than on modern leaves

We observed many more examples of DT297 on fossil than on modern leaves. The most striking difference is seen in *P. persica* and its fossil relative *P. pristina*, as we observed around 500 fossil specimens from Willershausen [26] versus more than 2300 modern specimens in its modern range in the Hyrcanian forest [133], and yet the large majority of DT297 has been observed on the fossil leaves (electronic supplementary material, S3). This is also true for the Chinese fossil leaf assemblages, which recorded almost 10 times more occurrences of DT297 than the modern *P. subaequalis* leaves from the Yixing forest area (electronic supplementary material, S3). This unexpected pattern can be explained in various ways.

These significant differences in occurrence could indicate an ecological change for this specialist plant–insect interaction. The populations of insects specialized on *Parrotia* could have significantly decreased during the last 15 Myr until they became relictual in the Hyrcanian and Yixing forests. The large climatic changes during the Miocene [93,134,135] and the introduction of glacial–interglacial cycles in the Quaternary [52,95] had a huge impact on numerous plant species' populations and their distributions [55,136–138], including *Parrotia* [139–141]. However, there are some examples of insect species that survived the glacial–interglacial cycles and recolonized the same area, such as the arctic–alpine insect species *Arcynopteryx dichroa* in the Central European highlands [142].

Sampling biases might also have caused the marked differences of occurrences between fossil and modern leaves. The modern leaves sampled represent only 1 or 2 years of leaf shedding, whereas fossil leaves may represent many years of leaf production and hence environmental variation, such as dry or wet years. Fossil leaves from Shanwang were recovered from several layers (19 sub-units in total) of diatomaceous sedimentary rocks [143] and represent a maximum of 3 Myr of elapsed time [31,66,69]. At Willershausen, fossils were collected from a clay pit and fossils from various layers were mixed. Based on lithological differences and contrasting fossil preservation, it is certain that the leaves from this outcrop represent many years of deposition [144,145]. Plant–insect interactions can significantly change from 1 year to another owing to a multitude of factors, such as climate seasonality [146–149]. Thus, insect feeding observed in the fossil record is generally more representative of the global herbivory pattern on *Parrotia* taxa than the observations made on modern leaf litter.

Only increased sampling efforts for *Parrotia* leaves from modern sites and fossil assemblages will enable better characterization of this differences in DT occurrences. Accordingly, one of the main objectives of our study was to thoroughly describe the specific DT DT297 in order to provide a basis for more comprehensive investigations in the future.

# 5. Conclusion

This study highlights and describes a long-term mode of herbivory, expressed as a skeletonization, which is exclusively represented on *Parrotia* species. We corroborate the damage (DT297) affinity to *Parrotia* taxa, for at least 15 Myr, and document this likeness relationship by providing structural similarities and detailed measurements supported by statistics. This specific DT provides direct evidence, quite rare in palaeoecology, of a long-term relationship between a plant species and its herbivore. DT297 is currently the most specific long-term herbivory trace identifiable on the same modern plant lineage. Henceforth, in order to better understand this interaction, fieldwork should be made in the Hyrcanian and/or Yixing forests in order to directly observe the insect species causing this distinct damage form.

The continuous presence of this DT over 15 Myr demonstrates a robust and long-term plant–herbivore association, and provides new evidence for a shared biogeographic history of the two host plants. This may have implications for improved understanding of phylogenetic relationships between the western Eurasian and East Asian host plant species.

Data accessibility. All data are accessible in the electronic supplementary material files of this manuscript.

Authors' contributions. J.-F.T. and T.W. together covered the publication fees, B.A., X.Z., and T.W. collected fossil and/or modern leaves; B.A. made the analysis; B.A. wrote the draft; B.W., J.-F.T., and T.W. contributed to the discussion. All authors approved the publication.

Competing interests. We declare we have no competing interests.

Funding. We did not receive any specific grants for this project but enjoyed support from the Deutsche Forschungsgemeinschaft (DFG) to visit the Willershausen fossil collections in some museums of Germany, and the Strategic Priority Research Program of the Chinese Academy of Sciences (grant no. XDB26000000) and the National Natural Science Foundation of China (grant no. 41688103) to work on this publication.

Acknowledgements. We would like to thank Dr Thomas Denk for sharing some *Parrotia* fossil specimens from the Willershausen outcrop stored in Naturhistoriska riksmuseet Stockholm and for his significant help in the vocabulary used in the manuscript. We want to thank Dr McLoughlin for his considerable support in improving our manuscript. Thank you also to Dr Thomas Bastelberger for sharing with us some *P. pristina* samples from his private fossils from Willershausen. Thanks are owing to Prof. Anurag Agrawal (Cornell University) for discussions about the topic of mutualism and herbivory. We also thank Dr Vincent Girard and Dr Conrad Labandeira for fruitful discussions, which helped improve our manuscript. Angela Nekes (Siegen University, Germany) helped with the German translation of an old manuscript about the Willershausen outcrop. Lina Leschner and Dr Alexander Gehler (Geowissenschaftliches Museum, Göttingen) provided the photography of the specimen *Parrotia* from Willershausen used as holotype for DT297.

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
