## [Reviewer comments · Royal Society Open Science]

Review History

RSOS-200244.R0 (Original submission)

Review form: Reviewer 1

Is the manuscript scientifically sound in its present form?

Yes

Are the interpretations and conclusions justified by the results?

Yes

Is the language acceptable?

No

Do you have any ethical concerns with this paper?

No

Have you any concerns about statistical analyses in this paper?

Yes

Recommendation?

Reject

Comments to the Author(s)

Review of RSOS-200244.

This manuscript has some merits in describing a very distinctive style of leaf herbivory that has remained unchanged for 15 million years and in that time has remained restricted to *Parrotia* species, albeit that both the representation of the plant genus and the herbivory style has declined across Eurasia during that time. The data is a useful case study illustrating 'evolutionary stasis' and also shows that not only can plant form remain very stable over long intervals, but host-herbivore interactions can be equally stable.

That said, this manuscript has some serious problems. Most significantly, the English grammar needs a great deal of attention. The phraseology is awkward throughout the text. Grammatical mistakes are plentiful. The names of genera are commonly not written in full at the beginnings of sentences. The formatting of references is inconsistent. US and UK spellings are intermixed. All these things detract from the scientific aspects of the paper. The reader becomes bogged down trying to untangle sentences rather than being enlightened about the topic. I began copy-editing the text but, after section 3.2.2, I gave up because the task would have required essentially re-writing the manuscript. I would recommend the authors have a native English speaker thoroughly edit the text before resubmission.

There are other issues with the manuscript:

The abstract/summary should not be an introduction. It should indicate the outcomes of the study rather than use nebulous phrases like "highlights new perspectives in the understanding of plant herbivory". - What are those new perspectives?

The text could be shortened in several places. There are some repetitive statements, some extended discussion that is not particularly relevant, and improvement to the grammar would condense sentences.

Avoid temporal terms (e.g., sometimes; occasionally; often; frequently) when describing fossils, as these imply that if you come back tomorrow, the fossils may show something entirely different. Better to use terms of generality (e.g., commonly; in some cases; generally, a few; many).

The term Tertiary is no longer used for a geological time period. This term should be replaced throughout the text.

Sections 4.1 and 4.2 both include measurements of damage dimensions. Why not merge these sections?

I do not have access to the paper by Straus A. 1977. (Gallen, Minen und andere Fraßspuren im Pliozän von Willershausen am Harz. Verhandlungen des Botanischen Vereins der Provinz Brandenburg. 113:41-80). From the abstract of this manuscript, the reader is led to understand that the trace fossils being described here are new to science. However, in section 4.5 we read that Straus 1977 already gave this damage a formal ichnotaxon name (*Phagophytichnus catellarius*) and designated a holotype. If this is correct, then it is not valid to call this a "new" damage type, nor is it appropriate to select a new 'holotype' for DT297. If the original holotype has been lost, then, at best, a lectotype should be chosen – otherwise the holotype remains!

A disappointment from the study is that no herbivore culprit has been identified. There can not be too many overlapping insect taxa between these widely segregated modern forests so it should not be too difficult to narrow down the herbivore.

Sections 5.2 to 5.4 could be condensed. Yes, it is interesting that the herbivory association has been so consistent and uniform for 15 million years, and the decline in both the representation of the plant and the intensity of herbivory is worth pointing out, but I don't think it requires a page

of single-spaced text to outline these issues. Moreover, the arguments about the role of 'mutualistic herbivory' or 'overcompensation' as outlined by Agrawal (2000) are hotly debated. The majority of the models dealing with this issue in modern plants are concerned with the release of apical dominance as the mechanism of overcompensation and come with several key assumptions relating to plant reproductive viability. Since this is almost impossible to test on the available fossils, much of section 5.4 is speculative.

The figures require some minor emendments. The reconstruction (Fig. 4) is well prepared.

Although the description of the herbivore damage indicates there is no consistent origin for the strands that separate the individual holes in the chain of damage features, several of the images in Fig. 3 seem to show that it is tertiary veins that represent the cross-struts within the damage chain.

Review form: Reviewer 2

Is the manuscript scientifically sound in its present form?

Yes

Are the interpretations and conclusions justified by the results?

No

Is the language acceptable?

No

Do you have any ethical concerns with this paper?

No

Have you any concerns about statistical analyses in this paper?

Yes

Recommendation?

Major revision is needed (please make suggestions in comments)

Comments to the Author(s)

I reviewed the manuscript entitled: A case of long-term herbivory: specialised feeding trace on Parrotia plant species by Adroit et al.

The manuscript is about a striking discovery of insect-plant interaction on two isolated relict trees in Iran and China. The authors accomplished a historical comparison of their finding using fossil materials of two stations in Willershausen and Shanwang, both representing historical occurrence of Parrotia species in an ancient time.

My general comment for improving the manuscript are as following:

1- We need to make sure whether this kind of damage could be also observed in other plants in the same modern habitats of both regions of Iran and China. How were your sampling frames in the Hyrcanian and Yixing forests? Have you observed enough leaves from other neighbor trees in your sampling frames? Is insect trace really specialist or generalist? these kinds of questions should be addressed in the method section.

2-First paragraph of the result (lines 147-157) should hold percentage values too. For example, you have 32 leaves of *P. persica* in Willershausen site (32 out of 517 Parrotia leaves in this site = ca. 6%). Then the ratio of damaged leaves in the Hyrcanian forest declines by 0.03 % (6 leaves out of 2160 leaves). Is it correct? Did you collect 2160 leaves of Parrotia in the Hyrcanian forest? If the damage ratio is really low (ca. 0.03 %), then one could expect to have same (or more or less same) ratio of infection for other species, IF other species had been hosting this damage. It means that

the authors should examine at least same number of leaves in other modern species in order to make sure that this damage is exclusively found on *Parrotia* leaves

3-Lines 32-33 of introduction indicates that the leaves of *P. persica* were the most abundant ones in the fossil collection. However, based on the figures you provided in the manuscript; this should be about 6% of all collection ($517/8073 = 6\%$)! Please also clarify the number 10,000 in line 31.

4-Is there any reason why you haven't included other two fossil sites already investigated in Adroit et al. 2018? Perhaps if you include more data, you would propose more robust statistical tests indicating a significant decline of damage through historical time.

Minor suggestions:

-The correct nomenclature for *Parrotia persica* is *Parrotia persica* (DC.) C. A. Mey. Please correct the author name of the species and try not to italic authors names in whole parts of the text.

-Scientific names in the abstract are written in upright letters. Is abstract an exception according to RSOS? Please check this.

Decision letter (RSOS-200244.R0)

01-Apr-2020

Dear Dr Adroit:

Manuscript ID RSOS-200244 entitled "A case of long-term herbivory: specialised feeding trace on *Parrotia* plant species" which you submitted to Royal Society Open Science, has been reviewed. The comments from reviewers are included at the bottom of this letter.

In view of the criticisms of the reviewers, the manuscript has been rejected in its current form. However, a new manuscript may be submitted which takes into consideration these comments.

Please note that resubmitting your manuscript does not guarantee eventual acceptance, and that your resubmission will be subject to peer review before a decision is made.

Your resubmitted manuscript should be submitted by 29-Sep-2020. If you are unable to submit by this date please contact the Editorial Office.

Kind regards,
Andrew Dunn
Senior Publishing Editor
Royal Society Open Science
openscience@royalsociety.org

on behalf of Professor Elizabeth Harper (Associate Editor)
openscience@royalsociety.org

Associate Editor Comments to Author (Professor Elizabeth Harper):

This is clearly an interesting study, however the reviewers have identified a number of important areas where the manuscript needs to be improved for it to be reviewed again. Some of these concerns reflect the basic science (including data gathering, comparisons with other interactions and ichnotaxonomy) but there are also major problems with the writing which requires substantial effort in order to allow the science be properly understood and evaluated.

Reviewers' Comments to Author:

Reviewer: 1

Comments to the Author(s)

Review of RSOS-200244.

This manuscript has some merits in describing a very distinctive style of leaf herbivory that has remained unchanged for 15 million years and in that time has remained restricted to *Parrotia* species, albeit that both the representation of the plant genus and the herbivory style has declined across Eurasia during that time. The data is a useful case study illustrating 'evolutionary stasis' and also shows that not only can plant form remain very stable over long intervals, but host-herbivore interactions can be equally stable.

That said, this manuscript has some serious problems. Most significantly, the English grammar needs a great deal of attention. The phraseology is awkward throughout the text. Grammatical mistakes are plentiful. The names of genera are commonly not written in full at the beginnings of sentences. The formatting of references is inconsistent. US and UK spellings are intermixed. All these things detract from the scientific aspects of the paper. The reader becomes bogged down trying to untangle sentences rather than being enlightened about the topic. I began copy-editing the text but, after section 3.2.2, I gave up because the task would have required essentially re-writing the manuscript. I would recommend the authors have a native English speaker thoroughly edit the text before resubmission.

There are other issues with the manuscript:

The abstract/summary should not be an introduction. It should indicate the outcomes of the study rather than use nebulous phrases like "highlights new perspectives in the understanding of plant herbivory". - What are those new perspectives?

The text could be shortened in several places. There are some repetitive statements, some extended discussion that is not particularly relevant, and improvement to the grammar would condense sentences.

Avoid temporal terms (e.g., sometimes; occasionally; often; frequently) when describing fossils, as these imply that if you come back tomorrow, the fossils may show something entirely different. Better to use terms of generality (e.g., commonly; in some cases; generally, a few; many).

The term Tertiary is no longer used for a geological time period. This term should be replaced throughout the text.

Sections 4.1 and 4.2 both include measurements of damage dimensions. Why not merge these sections?

I do not have access to the paper by Straus A. 1977. (Gallen, Minen und andere Fraßspuren im Pliozän von Willershausen am Harz. Verhandlungen des Botanischen Vereins der Provinz Brandenburg. 113:41-80). From the abstract of this manuscript, the reader is led to understand that the trace fossils being described here are new to science. However, in section 4.5 we read that

Straus 1977 already gave this damage a formal ichnotaxon name (*Phagophytichnus catellarius*) and designated a holotype. If this is correct, then it is not valid to call this a “new” damage type, nor is it appropriate to select a new ‘holotype’ for DT297. If the original holotype has been lost, then, at best, a lectotype should be chosen – otherwise the holotype remains!

A disappointment from the study is that no herbivore culprit has been identified. There can not be too many overlapping insect taxa between these widely segregated modern forests so it should not be too difficult to narrow down the herbivore.

Sections 5.2 to 5.4 could be condensed. Yes, it is interesting that the herbivory association has been so consistent and uniform for 15 million years, and the decline in both the representation of the plant and the intensity of herbivory is worth pointing out, but I don’t think it requires a page of single-spaced text to outline these issues. Moreover, the arguments about the role of ‘mutualistic herbivory’ or ‘overcompensation’ as outlined by Agrawal (2000) are hotly debated. The majority of the models dealing with this issue in modern plants are concerned with the release of apical dominance as the mechanism of overcompensation and come with several key assumptions relating to plant reproductive viability. Since this is almost impossible to test on the available fossils, much of section 5.4 is speculative.

The figures require some minor emendments. The reconstruction (Fig. 4) is well prepared.

Although the description of the herbivore damage indicates there is no consistent origin for the strands that separate the individual holes in the chain of damage features, several of the images in Fig. 3 seem to show that it is tertiary veins that represent the cross-struts within the damage chain.

Reviewer: 2

Comments to the Author(s)

I reviewed the manuscript entitled: A case of long-term herbivory: specialised feeding trace on *Parrotia* plant species by Adroit et al.

The manuscript is about a striking discovery of insect-plant interaction on two isolated relict trees in Iran and China. The authors accomplished a historical comparison of their finding using fossil materials of two stations in Willershausen and Shanwang, both representing historical occurrence of *Parrotia* species in an ancient time.

My general comment for improving the manuscript are as following:

1- We need to make sure whether this kind of damage could be also observed in other plants in the same modern habitats of both regions of Iran and China. How were your sampling frames in the Hyrcanian and Yixing forests? Have you observed enough leaves from other neighbor trees in your sampling frames? Is insect trace really specialist or generalist? these kinds of questions should be addressed in the method section.

2-First paragraph of the result (lines 147-157) should hold percentage values too. For example, you have 32 leaves of *P. persica* in Willershausen site (32 out of 517 *Parrotia* leaves in this site = ca. 6%). Then the ratio of damaged leaves in the Hyrcanian forest declines by 0.03 % (6 leaves out of 2160 leaves). Is it correct? Did you collect 2160 leaves of *Parrotia* in the Hyrcanian forest? If the damage ratio is really low (ca. 0.03 %), then one could expect to have same (or more or less same) ratio of infection for other species, IF other species had been hosting this damage. It means that the authors should examine at least same number of leaves in other modern species in order to make sure that this damage is exclusively found on *Parrotia* leaves

3-Lines 32-33 of introduction indicates that the leaves of *P. persica* were the most abundant ones in the fossil collection. However, based on the figures you provided in the manuscript; this should be about 6% of all collection ($517/8073 = 6\%$)! Please also clarify the number 10,000 in line 31.

4-Is there any reason why you haven’t included other two fossil sites already investigated in Adroit et al. 2018? Perhaps if you include more data, you would propose more robust statistical tests indicating a significant decline of damage through historical time.

Minor suggestions:

-The correct nomenclature for *Parrotia persica* is *Parrotia persica* (DC.) C. A. Mey. Please correct the author name of the species and try not to italic authors names in whole parts of the text.

-Scientific names in the abstract are written in upright letters. Is abstract an exception according to RSOS? Please check this.

Author's Response to Decision Letter for (RSOS-200244.R0)

See Appendix A.

RSOS-200737.R0

Review form: Reviewer 1

Is the manuscript scientifically sound in its present form?

No

Are the interpretations and conclusions justified by the results?

Yes

Is the language acceptable?

No

Do you have any ethical concerns with this paper?

No

Have you any concerns about statistical analyses in this paper?

No

Recommendation?

Major revision is needed (please make suggestions in comments)

Comments to the Author(s)

This manuscript is substantially improved from the earlier version, however, some issues remain. The phraseology still needs some tidying throughout the text, and there are many minor grammatical glitches and inconsistencies in the referencing style. I attach an annotated pdf (Appendix B) with additional suggestions for improvement to the grammar and comments on specific issues.

The last sentence of the 1st paragraph of the introduction does not hang together very well with the preceding text. It needs some connection between the sentences dealing with ecosystem balance / quantification of herbivory damage, and the last sentence dealing with historical establishment of a catalogue of damage types.

Throughout the text, the distinctive damage features on *Parrotia* species are described as “new”. This is misleading, as the damage was described in 1977 by Straus under a formal ichnospecies name. The only thing that is “new” about the illustrated damage is that it has a new designation “DT297” in a proposed future revision of the Labandeira et al. (2007) damage catalogue.

Throughout the text, the fossil *Parrotia* leaves are described as “analogues” of modern *Parrotia* plants. I think this is the wrong choice of words. They might have occupied analogous environments, but the plants themselves might be better described as “relatives”.

In section 3.1, the authors present a narrative of the “migration” of *Parrotia* across Eurasia-North America in the Cenozoic. Yet all the fossil record can really tell us is that a taxon is present at a particular place at a particular time. Absence of fossils does not necessarily mean absence of the taxon in the region – just that it hasn’t been preserved at that specific site. Moreover, many regions lack any fossil assemblages of the relevant age. In general, a narrative has been created based on a range of assumptions about how the plants responded to climate change, mountain belt uplift, etc. One can never really confirm the direction of “migration”/range expansion unless one has fossils from every year of the Cenozoic preserved in every region of the Northern Hemisphere.

There is some repetition between sections 3.1 and 3.2 – see attached pdf.

In section 4.1, the damage on *Parrotia* is described as a “mine”, yet it is also described as consisting of several “holes” or “chain of holes” (suggesting hole feeding, and in section 5.1 it is described as a form of “skeletonization”. There needs to be some consistency in the description and categorization of this damage type.

I would hope that, to maintain some consistency between formal ichnotaxonomic schemes and the informal damage categories employed by Labandeira et al. 2007, the holotype chosen for *Phagophytichnus catellarius* is the same specimen chosen as the reference specimen for DT297. If not, there may be issues in the future about how these damage features are named.

The authors need to be careful with their use of Ma and Myrs. Ma is used to denote a point in time (20 Ma = 20 million years ago), whereas Myrs describes a span of time (20 Myrs = a 20-million-year interval).

The figures look okay, but the captions need some rephrasing.

Despite the changes still required, the manuscript is useful in describing a very distinctive style of leaf herbivory that has remained unchanged for 15 million years, and which has remained restricted to *Parrotia* species. This is a useful case study illustrating ‘evolutionary stasis’ in host-herbivore interactions over a long interval.

Review form: Reviewer 2

Is the manuscript scientifically sound in its present form?

Yes

Are the interpretations and conclusions justified by the results?

Yes

Is the language acceptable?

No

Do you have any ethical concerns with this paper?

No

Have you any concerns about statistical analyses in this paper?

No

Recommendation?

Major revision is needed (please make suggestions in comments)

Comments to the Author(s)

Dear Authors,

The resubmitted version of the manuscript sounds better, still there are some weak points in the paper.

-English of the text is very weak. There are too many grammatical errata. I strongly suggest that a native speaker needs to read the text before resubmission.

- What is sampling frame mentioned in line 78? You should explain more how you did your sampling. Did you collect leaves on a random-based framework? How many sites/locations were assessed during your sampling? How many leaves of other species have been included in your sampling? This phenomenon is very rare in the modern leaves (as you already insisted in your MS), then one could hypothesize that the same herbivory could also probably happen on other trees like Zelkova, Alnus, Fagus or Carpinus if enormous number of leaves of the latter species would be collected in the sampling plots. Please

-I am still not very confident that herbivory trace on fossil and modern leaves are really coming from same type of herbivore. Could you discuss more about other similar type of traces?

-I can not see any reason to include lines 218 to 222 in the discussion part. This is part of your result.

Decision letter (RSOS-200737.R0)

Dear Dr Adroit:

Manuscript ID RSOS-200737 entitled "A case of long-term herbivory: specialised feeding trace on Parrotia plant species" which you submitted to Royal Society Open Science, has been reviewed. The comments from reviewer(s) are included at the bottom of this letter.

In view of the criticisms of the reviewer(s), I must decline the manuscript for publication in Royal Society Open Science at this time. However, a new manuscript may be submitted which takes into consideration these comments. One of the reviewers has provided a PDF to assist you. As the ScholarOne system sometimes 'scrubs' large attachments from emails, please contact the editorial office if you have any difficulty accessing the PDF, and they will send it to you directly.

Please note that resubmitting your manuscript does not guarantee eventual acceptance, and that your resubmission will be subject to re-review by the reviewer(s) before a decision is rendered.

You will be unable to make your revisions on the originally submitted version of your manuscript. Instead, revise your manuscript using a word processing program and save it on your computer.

You may also click the below link to start the resubmission process (or continue the process if you have already started your resubmission) for your manuscript. If you use the below link you will not be required to login to ScholarOne Manuscripts.

*** PLEASE NOTE: This is a two-step process. After clicking on the link, you will be directed to a webpage to confirm. ***

https://mc.manuscriptcentral.com/rsos?URL_MASK=d6549f923458483c979268d35448c572

Because we are trying to facilitate timely publication of manuscripts submitted to Royal Society Open Science, your resubmitted manuscript should be submitted by 11-Dec-2020. If you are unable to submit by this date please contact the Editorial Office for options.

I look forward to a resubmission.

on behalf of Professor Elizabeth Harper (Associate Editor)
openscience@royalsociety.org

Associate Editor Comments to Author (Professor Elizabeth Harper):

This is a very interesting study and it is great to see that this manuscript has substantially improved and both reviewers acknowledge this. However, there are still major issues in particular those highlighted by McLoughlin. These need to be thoroughly addressed in full before any resubmission.

Reviewer comments to Author:

Reviewer: 1

Comments to the Author(s)

This manuscript is substantially improved from the earlier version, however, some issues remain. The phraseology still needs some tidying throughout the text, and there are many minor grammatical glitches and inconsistencies in the referencing style. I attach an annotated pdf with additional suggestions for improvement to the grammar and comments on specific issues.

The last sentence of the 1st paragraph of the introduction does not hang together very well with the preceding text. It needs some connection between the sentences dealing with ecosystem balance / quantification of herbivory damage, and the last sentence dealing with historical establishment of a catalogue of damage types.

Throughout the text, the distinctive damage features on *Parrotia* species are described as "new". This is misleading, as the damage was described in 1977 by Straus under a formal ichnospecies name. The only thing that is "new" about the illustrated damage is that it has a new designation "DT297" in a proposed future revision of the Labandeira et al. (2007) damage catalogue.

Throughout the text, the fossil *Parrotia* leaves are described as "analogues" of modern *Parrotia* plants. I think this is the wrong choice of words. They might have occupied analogous environments, but the plants themselves might be better described as "relatives".

In section 3.1, the authors present a narrative of the "migration" of *Parrotia* across Eurasia-North America in the Cenozoic. Yet all the fossil record can really tell us is that a taxon is present at a particular place at a particular time. Absence of fossils does not necessarily mean absence of the taxon in the region - just that it hasn't been preserved at that specific site. Moreover, many regions lack any fossil assemblages of the relevant age. In general, a narrative has been created

based on a range of assumptions about how the plants responded to climate change, mountain belt uplift, etc. One can never really confirm the direction of “migration”/range expansion unless one has fossils from every year of the Cenozoic preserved in every region of the Northern Hemisphere.

There is some repetition between sections 3.1 and 3.2 – see attached pdf.

In section 4.1, the damage on *Parrotia* is described as a “mine”, yet it is also described as consisting of several “holes” or “chain of holes” (suggesting hole feeding, and in section 5.1 it is described as a form of “skeletonization”. There needs to be some consistency in the description and categorization of this damage type.

I would hope that, to maintain some consistency between formal ichnotaxonomic schemes and the informal damage categories employed by Labandeira et al. 2007, the holotype chosen for *Phagophytichnus catellarius* is the same specimen chosen as the reference specimen for DT297. If not, there may be issues in the future about how these damage features are named.

The authors need to be careful with their use of Ma and Myrs. Ma is used to denote a point in time (20 Ma = 20 million years ago), whereas Myrs describes a span of time (20 Myrs = a 20-million-year interval).

The figures look okay, but the captions need some rephrasing.

Despite the changes still required, the manuscript is useful in describing a very distinctive style of leaf herbivory that has remained unchanged for 15 million years, and which has remained restricted to *Parrotia* species. This is a useful case study illustrating ‘evolutionary stasis’ in host-herbivore interactions over a long interval.

Reviewer: 2

Comments to the Author(s)

Dear Authors,

The resubmitted version of the manuscript sounds better, still there are some weak points in the paper.

-English of the text is very weak. There are too many grammatical errata. I strongly suggest that a native speaker needs to read the text before resubmission.

- What is sampling frame mentioned in line 78? You should explain more how you did your sampling. Did you collect leave on a random-based framework? How many sites/locations were assessed during your sampling? How many leaves of other species have been included in your sampling? This phenomenon is very rare in the modern leaves (as you already insisted in your MS), then one could hypothesize that the same herbivory could also probably happen on other trees like *Zelkova*, *Alnus*, *Fagus* or *Carpinus* if enormous number of leaves of the latter species would be collected in the sampling plots. Please

-I am still not very confident that herbivory trace on fossil and modern leaves are really coming from same type of herbivore. Could you discuss more about other similar type of traces?

-I can not see any reason to include lines 218 to 222 in the discussion part. This is part of your result.

Author's Response to Decision Letter for (RSOS-200737.R0)

See Appendix C.

RSOS-201449.R0

Review form: Reviewer 2

Is the manuscript scientifically sound in its present form?

Yes

Are the interpretations and conclusions justified by the results?

Yes

Is the language acceptable?

Yes

Do you have any ethical concerns with this paper?

No

Have you any concerns about statistical analyses in this paper?

No

Recommendation?

Accept as is

Comments to the Author(s)

I read the manuscript along with responses provided by the authors and I found that the current version is suitable for publication

Decision letter (RSOS-201449.R0)

Dear Dr Adroit

On behalf of the Editors, we are pleased to inform you that your Manuscript RSOS-201449 "A case of long-term herbivory: specialised feeding trace on *Parrotia* (Hamamelidaceae) plant species" has been accepted for publication in Royal Society Open Science subject to minor revision in accordance with the referees' reports. Please find the referees' comments along with any feedback from the Editors below my signature.

We invite you to respond to the comments and revise your manuscript. Below the referees' and Editors' comments (where applicable) we provide additional requirements. Final acceptance of your manuscript is dependent on these requirements being met. We provide guidance below to help you prepare your revision. Please ensure you seek additional linguistic guidance from a service such as those at <https://royalsociety.org/journals/authors/benefits/language-editing/> or a native speaker of English to iron out the remaining wrinkles with the text.

Please submit your revised manuscript and required files (see below) no later than 7 days from today's (ie 15-Sep-2020) date. Note: the ScholarOne system will 'lock' if submission of the revision

is attempted 7 or more days after the deadline. If you do not think you will be able to meet this deadline please contact the editorial office immediately.

on behalf of Professor Elizabeth Harper (Associate Editor)
openscience@royalsociety.org

Associate Editor Comments to Author (Professor Elizabeth Harper):

Thank you for your thorough revision. The reviews have now been answered well and this, I believe, will be a very interesting paper. However, since Dr McLoughlin has been so helpful in improving this paper I would like to see his efforts acknowledged at the end of the paper.

Reviewer comments to Author:
Reviewer: 2

Comments to the Author(s)

I read the manuscript along with responses provided by the authors and I found that the current version is suitable for publication

===PREPARING YOUR MANUSCRIPT===

- one version identifying all the changes that have been made (for instance, in coloured highlight, in bold text, or tracked changes);
- a 'clean' version of the new manuscript that incorporates the changes made, but does not highlight them.

This version will be used for typesetting.

===PREPARING YOUR REVISION IN SCHOLARONE===

Author's Response to Decision Letter for (RSOS-201449.R0)

See Appendix D.

Decision letter (RSOS-201449.R1)

Dear Dr Adroit,

It is a pleasure to accept your manuscript entitled "A case of long-term herbivory: specialised feeding trace on *Parrotia* (Hamamelidaceae) plant species" in its current form for publication in Royal Society Open Science.

on behalf of Professor Elizabeth Harper (Associate Editor) and Andrew Dunn (Subject Editor)
openscience@royalsociety.org

Appendix A

Dear Prof Elizabeth Harper,

We carefully read all the comments and took them into account to review our manuscript. English has been fully corrected by native English speaker.

After several discussions with authors and researchers involved in this topic, we reconsidered the ichnotaxonomy of the damage.

We added a small part in both M&M and discussion which are link to the plant taxonomy determination of the genus *Parrotia* and the inputs highlighted by this plant-insect interaction.

We hope that you will appreciate the reviewed version of our manuscript.

Sincerely yours,

Dr. Benjamin Adroit.

REVIEWER 1

The abstract/summary should not be an introduction. It should indicate the outcomes of the study rather than use nebulous phrases like “highlights new perspectives in the understanding of plant herbivory”. - What are those new perspectives?

We corrected the abstract following these recommendations.

The text could be shortened in several places. There are some repetitive statements, some extended discussion that is not particularly relevant, and improvement to the grammar would condense sentences.

Avoid temporal terms (e.g., sometimes; occasionally; often; frequently) when describing fossils, as these imply that if you come back tomorrow, the fossils may show something entirely different. Better to use terms of generality (e.g., commonly; in some cases; generally, a few; many).

All the manuscript has been reviewed by English native speaker.

The term Tertiary is no longer used for a geological time period. This term should be replaced throughout the text.

We used the term Cenozoic in all the manuscript. Only the term “Artco-Tertiary plant species” remains as it is the most common term in the scientific literature to mean the “geoflora” assemblage that covered Northern Hemisphere during the Cenozoic. “Artco-Cenozoic flora” is not a term used in scientific literature.

Sections 4.1 and 4.2 both include measurements of damage dimensions. Why not merge these sections?

We merged both sections together.

I do not have access to the paper by Straus A. 1977. (Gallen, Minen und andere Fraßspuren im Pliozän von Willershäusen am Harz. Verhandlungen des Botanischen Vereins der Provinz Brandenburg. 113:41-80). From the abstract of this manuscript, the reader is led to understand that the trace fossils being described here are new to science. However, in

section 4.5 we read that Straus 1977 already gave this damage a formal ichnotaxon name (*Phagophytichnus catellarius*) and designated a holotype. If this is correct, then it is not valid to call this a “new” damage type, nor is it appropriate to select a new ‘holotype’ for DT297. If the original holotype has been lost, then, at best, a lectotype should be chosen – otherwise the holotype remains!

We reconsidered the nomenclature for the new DT297. To be fair with scientific requirements, we prefer to not talk about neither holotype nor lectotype, paratype etc. We simply call as a “reference” to make it easier for the scientific literature.

A disappointment from the study is that no herbivore culprit has been identified. There can not be too many overlapping insect taxa between these widely segregated modern forests so it should not be too difficult to narrow down the herbivore.

We do not share the point of view of the reviewer for that point.

If it was easy to determine some plausible insects, then we would do it. The reviewer wrote later in our manuscript that one paragraph was too much speculative and this is actually what this comment suggests to do here: to speculate.

We mentioned a plausible insect genus which could did this damage according to our personal observations, knowledges, biogeographic comparisons, discussions with entomologists and hypothesis from Strauss.

Sections 5.2 to 5.4 could be condensed.

We reduced those part and merged them in one.

Yes, it is interesting that the herbivory association has been so consistent and uniform for 15 million years, and the decline in both the representation of the plant and the intensity of herbivory is worth pointing out, but I don’t think it requires a page of single-spaced text to outline these issues. Moreover, the arguments about the role of ‘mutualistic herbivory’ or ‘overcompensation’ as outlined by Agrawal (2000) are hotly debated. The majority of the models dealing with this issue in modern plants are concerned with the release of apical dominance as the mechanism of overcompensation and come with several key assumptions relating to plant reproductive viability. Since this is almost impossible to test on the available fossils, much of section 5.4 is speculative.

We reduced this part in our manuscript.

Agrawal hypothesis cannot test the temporal perspective of the plausible mutualistic relation between insect herbivore and plant host. The point that we wanted to highlight is that this long-term herbivory damage could partially fill this gap.

The figures require some minor emendments.

We corrected them according to reviewer comments.

Although the description of the herbivore damage indicates there is no consistent origin for the strands that separate the individual holes in the chain of damage features, several of the images in Fig. 3 seem to show that it is tertiary veins that represent the cross-struts within the damage chain.

We changed description of the damage and classified as Skeletonization instead of hole feeding after discussions with Prof Conrad Labandeira.

REVIEWER 2

1- We need to make sure whether this kind of damage could be also observed in other plants in the same modern habitats of both regions of Iran and China. How were your sampling frames in the Hyrcanian and Yixing forests? Have you observed enough leaves from other neighbor trees in your sampling frames? Is insect trace really specialist or generalist? these kinds of questions should be addressed in the method section.

We rephrased the whole paragraph 5.1 which could be confusing according to those questions.

Only the fossil observations, according to the method of Labandeira et al., 2007, allow to classified DT297 as a “specialist” damage.

The modern observations are here to demonstrate that the DT297 is still on *Parrotia* plant species.

2-First paragraph of the result (lines 147-157) should hold percentage values too. For example, you have 32 leaves of *P. persica* in Willershausen site (32 out of 517 *Parrotia* leaves in this site = ca. 6%). Then the ratio of damaged leaves in the Hyrcanian forest declines by 0.03 % (6 leaves out of 2160 leaves). Is it correct? Did you collect 2160 leaves of *Parrotia* in the Hyrcanian forest? If the damage ratio is really low (ca. 0.03 %), then one could expect to have same (or more or less same) ratio of infection for other species, IF other species had been hosting this damage. It means that the authors should examine at least same number of leaves in other modern species in order to make sure that this damage is exclusively found on *Parrotia* leaves

A part of the answer is include in the previous comment.

Also, we mentioned the % according to the whole samples of *Parrotia* leaves and then we state that we observe more occurrence in the fossil records, and many reasons can explain this difference, we wanted to suggest them.

The % proposed by the reviewer would be to estimate the ratio of the damage DT297 for each population of *Parrotia* species. We think that we cannot pretend to discuss about this % comparison, as the quantity of leaves measured in the Chinese fossil outcrop and Chinese modern forest (due to the low quantity of trees) are too low to be compared with the Hyrcanian forest for example.

The % ratio of DT297 per *Parrotia* leaf in Chinese locations could be too much overstate. Actually, our manuscript never pretends to compare the proportion of the damage DT297 per *Parrotia* populations as it would not be a fair and robust comparison. The main statement is to show and describe a new damage type on *Parrotia* genus.

3-Lines 32-33 of introduction indicates that the leaves of *P. persica* were the most abundant ones in the fossil collection. However, based on the figures you provided in the manuscript; this should be about 6% of all collection ($517/8073 = 6\%$)!

Yes, 517 leaves is the maximum number of leaves from a same plant species in the fossil collection of Willershausen. Those 517 fossil leaves are of *Parrotia persica* (= *P. pristina*).

Then *P. persica* is the most abundant species represented in the macroflora of Willershausen fossil leaf collection analyzed in Adroit et al., 2018a.

Please also clarify the number 10,000 in line 31.

We rephrased the sentence to be more accurate.

4-Is there any reason why you haven't included other two fossil sites already investigated in Adroit et al. 2018? Perhaps if you include more data, you would propose more robust statistical tests indicating a significant decline of damage through historical time.

We clearly added in the discussion that Bernasso and Berga did not present any traces of DT297 on *P. persica* leaves.

Minor suggestions:

-The correct nomenclature for *Parrotia persica* is *Parrotia persica* (DC.) C. A. Mey. Please correct the author name of the species and try not to italic authors names in whole parts of the text.

We corrected it.

-Scientific names in the abstract are written in upright letters. Is abstract an exception according to RSOS? Please check this.

We corrected it.

A case of long-term herbivory: specialised feeding trace on *Parrotia* (Hamamelidaceae) plant species

Benjamin Adroit^{a*}, Xin Zhuang^b, Torsten Wappler^c, Jean-Frederic Terral^d, Bo Wang^a

a State Key Laboratory of Palaeobiology and Stratigraphy, Nanjing Institute of Geology and Palaeontology, Chinese Academy of Sciences, 39 East Beijing Road, Nanjing 210008, China

b College of Life Sciences, Nanjing University, 22 Hankou Road, Nanjing 210093, China

c Hessisches Landesmuseum Darmstadt, Darmstadt, Germany

d Institut des Sciences de l'Evolution, UMR5554 Université de Montpellier, CNRS, IRD, EPHE, Place Eugène Bataillon, 34095 Montpellier, cedex 05, France

Keywords: palaeoecology, plant-insect interaction, fossil leaf, endemism

1. Summary

Interactions between plants and insects evolved during millions of years of co-evolution and maintain the trophic balance of terrestrial ecosystems. Documenting insect damage types on fossil leaves is essential for understanding the evolution of plant-insect interactions and for understanding the effects of major environmental changes on ecosystem structure. However, research focusing on herbivory is still sparse and only a tiny fraction of fossil leaf collections has been analysed. This study documents a new type of insect damage found exclusively on the leaves of *Parrotia* species (Hamamelidaceae). This new damage type was identified on *Parrotia* leaves from Willershausen (Germany, Pliocene) and from Shanwang (China, Miocene) and on their respective endemic modern analogues: *Parrotia perisca* in the Hyrcanian forests (Iran) and *Parrotia subaequalis* in the Yixing forest (China). Our study demonstrates that this insect damage type persisted over at least 15 million years spanning from eastern Asia to western Europe. Against expectations, more damage type occurrences were identified on the fossil leaves than on the modern examples. This mismatch may suggest a decline of this specialised plant-insect interaction due to the contraction of *Parrotia* populations in Eurasia during the late Cenozoic. However, the continuous presence of this damage type demonstrates a robust and long-term plant-herbivore association, and provides new evidence for a shared biogeographic history of the two host plants.

2. Introduction

An ecosystem is a set of abiotic environmental conditions with communities of organisms living and interacting therein. Interactions between organisms contribute to a fragile equilibrium within ecosystems (1–3), and are major in the ecosystemic balance which is considerably disturbed by modern human activities (4–6). This global change is expected to affect plant-insect associations in different ways as insects play significant roles such as herbivores on plants (7) and pollinators for crop production (8). These interactions are the result of millions of years of evolution (9–11). Plant-herbivore interactions are of particular importance for terrestrial food webs that sustain biodiversity and ecosystem balance (12–14). Variation in the style and quantity of herbivory depends mostly on abiotic parameters, primarily climatic conditions (9,15–17). Consequently, it is not surprising that many studies have measured significant changes in the patterns of herbivory on fossil leaves through geological time as environmental conditions have changed (16,18–25). In order to standardize the identification of herbivory traces in scientific publications, Labandeira et al. (2007) (26) developed the “Guide to insect (and other) damage types on compressed plant fossils”.

*Author for correspondence, Benjamin Adroit (benjamin.adroit@gmail.com).

†Present address: Nanjing Institute of Geology and Palaeontology, No.39, East Beijing Road, Nanjing 210008, CHINA

These sentences don't hang together well! The last sentence lacks a clear connection to the rest of the paragraph. — and it is somewhat repeated in section 3.5
<https://mc.manuscriptcentral.com/rsos>

Later in the text, you say that this damage is not new at all, but was originally described in 1977 by Straus. Fix throughout text.

same thing

palaeo

examples of this type of herbivory

environmental

can be drastically

various

Recent research has been carried out on some late Cenozoic floras from Europe using standardized damage type nomenclature. This research includes work on the famous Lagerstätte of Willershausen in Germany (27). During the identification of herbivory traces on fossil leaves of Willershausen (i.e. 8,073 fossil specimens analysed), a new insect damage type was recorded exclusively ~~found~~ ^{similar to} on the fossil leaves analogue of *Parrotia persica* (DC.) C. A. Mey. These leaves were the most abundant within the fossil assemblage (27,28) and hosted many examples of this new insect feeding trace (27). Parallel to this study, another investigation was conducted mostly on modern leaves of *P. persica* from the Hyrcanian forest region (29). Nowadays, *P. persica* is endemic to the Hyrcanian forest region south of the Caspian Sea, together with many other Neogene relict species (30,31). During this study, herbivory traces similar to those on the fossils were observed on living *P. persica* at the locations Aliabad-e Katul, Pasand and Molla Kala in Iran (Figure 1). Lastly, within the framework of another project on fossil plants from the Lagerstätte of Shanwang (Miocene), north-east China (32,33), around 1,300 leaves were studied and the same insect feeding trace was identified on just one *Parrotia* leaf.

This study describes the type of the new plant-herbivore interaction found exclusively on the leaves of the two known *Parrotia* species. In addition to the fossil and modern materials mentioned previously, some modern leaves of *P. subaequalis* were also measured in the Yixing forest (China), where one of the small endemic populations of *P. subaequalis* still exists (34). We discuss how such external leaf feeding remained unchanged for 15 Ma across Eurasia, in the face of major environmental changes. Finally, we discuss how this discovery provides new perspectives on the evolution of plant-insect interactions.

3. Materials and Methods

In this study, modern leaf specimens of the genus *Parrotia* and their fossil analogues were studied.

3.1 Fossil record of *Parrotia*

Parrotia was present in East Asia and possibly North America during the Eocene and migrated to western Eurasia across Central Asia during the Oligocene (35). During the early Oligocene the genus was present in Kazakhstan (from where it disappeared during the Miocene) (36). From Europe, *Parrotia pristina* (Ettingshausen) Stur and *P. fugifolia* (Göppert) Heer were described. These names were also used for Paleogene and Neogene leaf fossils of East Asia (37,38). In addition, *Fothergilla* Hu and Chaney was described from the early to middle Miocene Shanwang flora of China (39). Based on the morphological similarity with the extant *Shaniodendron subaequalis* (= *Parrotia subaequalis*), *F. virburnifolia* was later transferred to *Shaniodendron virburnifolium* (Hu et Chaney) Wang et Li (40). In Europe and Kazakhstan, fossils assigned to *Parrotia* are commonly called *P. pristina* (*P. fugifolia* being a junior synonym). In East Asia, the nomenclature is somewhat unclear: *Shaniodendron virburnifolium* should be treated as *Parrotia virburnifolia* based on the current taxonomic treatment of *Shaniodendron* as a synonym of *Parrotia* (41). At the same time, this name competes with the earlier name *P. pristina*, which also has been used for East Asian examples, as *P. virburnifolia*.

3.2 *Parrotia persica* (DC.) C. A. Mey

Parrotia persica is a deciduous tree 8-25m tall (42). The leaves are oblong to obovate, up to 15 cm long and 6 cm wide, with 5-8 pairs of secondary veins (28,42,43). Nowadays, *P. persica* exists only in the Hyrcanian forest south of the Caspian Sea (Iran, Azerbaijan). The leaf shape of *P. persica* is very consistent despite there being size variability in leaves throughout Hyrcanian forest owing to variable local abiotic conditions (44). *Parrotia persica* is an Arcto-Tertiary relict species (35). Its close fossil relatives were very common in European forests during the Neogene (43,45-51). In this study, fossil leaves from Willershausen (Pliocene) in Germany and modern leaves from the Hyrcanian forest in northern Iran were comparatively studied (Figure 1).

3.2.1 Willershausen, 3 Ma, Germany

Willershausen is a Lagerstätte in the centre of Germany, close to Göttingen (Figure 1). It is a lacustrine clay pit containing more than 130 fossil plant species including many leaves of *Zelkova zelkovifolia*, *Carpinus orientalis* Mill., *Carya minor* Schenk and *P. pristina* (28,49,52), which is the most abundant in the fossil plant assemblage (27). The palaeoforest represented by this fossil leaf assemblage is dated around 3 Ma; MN 16/17 (35,53). Adroit et al. (2018) (27), analysed 8,073 leaf specimens of which 517 were attributed to fossil analogue of *Parrotia*.

Or at least you don't find any more fossils of it after the Miocene but it may still have been present.

Avoid single sentence paragraphs

what age?

Not really analogues - they are relatives to the extant species.

But how can you tell its direction of migration - you may not have fossils from its full temperate & geographic range.

Repetition

en dash, not hyphen

Clarify

relative? / representative?

3.2.2 Hyrcanian forest, modern, northern Iran

In terms of plant species richness, the Hyrcanian forest region (Figure 1) is considered a good modern analogue of the European forests of the late Cenozoic (43,54), such as that represented by the Willershausen Lagerstätte. The Hyrcanian forest region is a refuge ^{for} to several ^{extant} Arcto-Tertiary plant species ^{that} which are endemic to this area today (such as *P. persica*) (35,43,55,56). This forest extends from Golestan National Park (northeastern Iran) to eastern Azerbaijan, and is bordered by the Caspian Sea to the north and the Alborz mountains to the south, encompassing 1.85 million ha (31). Adroit et al. (2018b) (29) collected and analysed 2,160 leaves of *P. persica* and examined additional leaves from other species ^{commonly} (such as *Zelkova carpinifolia* (Pall.) K.Koch, *Quercus castaneifolia* C. A. Mey, *Acer cappadocicum* Gled.) which were frequently co-occurring with *Parrotia*.

3.3 *Parrotia subaequalis* (H.T.Chang) R.M.Hao & H.T.Wei

Similar to its sibling species (57,58), *P. subaequalis* is a large shrub or small tree, 5–10 m tall (57). Rarely, it reaches up to 20 m tall with pruning and staking, as evident in a village on Qingliang Peak, Linan, China. Leaf blades of *P. subaequalis* are mostly broad-obovate or elliptic, 4–6.5 cm long and 2–4.5 cm wide, ^{and} thinly leathery (59). *Parrotia subaequalis* is a Cenozoic relic plant species endemic to eastern China (60). Fossil specimens from Miocene strata indicate the former distribution of *Parrotia* in Shanwang, Shandong Province, northeastern Central China (61,62) and in Huadian, Jilin Province, northeastern China (63). Its population size severely decreased during Quaternary glaciations (64); the ~~lineage leading to the~~ modern species has a narrow and scattered distribution on Mt. Qinling-Dabie and Mt. Tianmu (China). *Parrotia subaequalis* was described from Yixing, Jiangsu province as *Hamamelis subaequalis* H.T. Chang and later transferred to the monotypic genus *Shaniodendron* (65). ~~Finally, observation of the flowers (61):~~ ^{en dash} Subsequently, flower morphology (61) and a molecular phylogenetic study (58) ~~Close relationship between *Shaniodendron* and *Parrotia*~~ suggested that *Shaniodendron* should be included within *Parrotia* resulting in the name *Parrotia subaequalis*. The modern leaves of *P. subaequalis* came from the Yixing forest in eastern China (Figure 1).

3.3.1 Shanwang, 18–15 Ma, China

Shanwang is a Lagerstätte (32) containing a diverse assemblage of organisms dominated by angiosperms (32,33). It is located in northeastern China, in Shandong province (Figure 1).

According to various dating methods, the Shanwang deposit is early-middle Miocene (66), i.e. 18–15 Ma (32,67–70). Both pollen and fossil leaf studies indicate the presence of *Quercus*, *Pterocarya*, *Ulmus*, *Populus*, *Fraxinus*, *Carpinus* and *Betula* (71). They represent an evergreen broad-leaved and mixed deciduous forest (71). The fossil collection from this deposit is stored in the Nanjing Institute of Geology and Palaeontology (Nanjing, China) and includes 1,298 leaves, of which 40 are attributed to *Parrotia*.

3.3.2 Yixing forest, modern, eastern China

The modern Yangtze River valley is an appropriate environmental analogue of the Shanwang Miocene site, although the Shanwang paleo-forest may have experienced lower annual temperatures including possibly colder summers and lower seasonality in rainfall. Yixing Forest Farm, located in the Yangtze River valley (Figure 1), is one of the most significant state-owned forest farms in southwestern Jiangsu Province, covering 34 km², with 97% forest coverage. This farm is set in the region of Mt. Yili, which is geographically a low-altitude hilly terrain ^{forming} at the east ^{ern} extension of Mt. Qinlin-Dabie (72). There is a small population of *P. subaequalis* trees in the central part of Yixing Forest Farm, with three eminent old trees and around 20 mature individuals. Other small populations occur within and around the farm (60). In the field, 41 leaves of *P. subaequalis* were sampled.

3.4 Observation

All the specimens were ^{studied} observed with a stereomicroscope (Leica EZ4) and a microscope (Zeiss AXIO Zoom V.16). They were photographed with a Lumix GX8 mounted on a copy stand. ^{transmitted light??} All the fossil leaves from the ^{various} different collections were sampled many years ago and described in previous works. No additional sampling was attempted, as the fossil collections are large enough and because, nowadays, collecting in Willershausen (Germany) and Shanwang (China) is forbidden. All the modern leaves were sampled from the ground (litter) in the Hyrcanian forest region (Iran) and Yixing forest (China). Leaves from the litter are more representative for plant-insect interactions as herbivory is not homogeneously distributed throughout the tree and the whole spectrum of leaf damage types is best captured when leaves from all parts of the tree including the canopy are considered (73–76). Moreover, to collect a fallen leaf from the litter means collecting after the whole lifespan of this leaf, and then no more herbivory can happen. Last ^y, the leaves from the litter represent at least a part of the taphonomy process. For those reasons, the leaf litter is better for standardization of samples for the whole study.

3.5 Terminology

Currently, the main reference to identify and classify the plant-insect interactions in the fossil record is the "Guide to Insect (and Other) Damage Types on Compressed Plant Fossils" (26). This guide subdivides herbivory traces into seven functional feeding groups (FFG) ^{on leaves} such as: hole feeding, margin feeding, skeletonization, surface feeding, mining, piercing & sucking and galling. For each FFG, numerous damage types (DT) are recognised. For each of these DTs, a host specificity index (HS) ^{is assigned that distinguishes} that allows to make a distinction between generalist and specialist damage (77). The determination of this HS index is based on diverse parameters, such as its geographical distribution, plant species diversity affected by the damage, damage quantity, shape variations, ^{among other factors} etc. (more details in 25,69).

3.6 Deposition of fossil specimens

Fossil *P. persica* leaves from Willershausen (Germany) analysed in this study are all deposited at the Geoscience Centre of the University of Göttingen (GZG.W collection). The fossil leaves of *P. viburnifolia* (labelled as *P. subaequalis* in the collection) belong to ^{the} Nanjing Institute of Geology and Palaeontology, Chinese Academy of Sciences (China).

3.7 Measurements

With the help of photography and the software ImageJ, each leaf was measured following several parameters, such as length, width and surface area of the leaf blade. ^{where} When it has been possible, the width of the petiole ^{has} also been measured in order to determine the Leaf Mass per Area (LMA) for each specimen. LMA is an index ^{that} which corresponds to the relation between the thickness and the density of the leaf (78,79). Thereafter, the specific damage on each *Parrotia* leaves was recorded and described, and the measurements of the surface area of the damage, the length and the width at three different positions along the damage, and the number of holes, were compared between the two *Parrotia* species and both fossil and modern leaves. Basic statistical tests based on the averages of measurements (Shapiro, Fisher and Wilcoxon) were made in addition to these morphological comparisons.

4. Results

At Willershausen, 32 leaves of *P. pristina* had the new damage type DT297 and we counted 143 occurrences in total. In general, one leaf can include more than one damage ^{occurrence} (Figure 2). In Shanwang, four leaves of the species *P. viburnifolia* (= *P. subaequalis* fossil analogue) had 13 occurrences of this specific damage type. In all of the fossil collections, no leaves from other species had ^{the} DT297. In the modern Hyrcanian forest, despite the large amount of *P. persica* leaves collected, only six had this new damage type, and in a low quantity as only seven occurrences in total ^{were recognized}. In the Yixing forest, ^{damage abundance} it is even lower; only four occurrences on two leaves of *P. subaequalis* were identified.

Overall, 167 damages ^{occurrences were} have been observed in this study among 43 leaves of *Parrotia* species (Figure 2). A large majority of them ^{were} have been observed on *P. pristina* ^{from} Willershausen (85%) and then 7% on *P. viburnifolia* from Shanwang. The modern samples of *Parrotia* spp. (i.e. *P. persica* and *P. subaequalis* together) represent 8% of our observations.

4.1 Morphological description

The average has been calculated by ^{scoring/counting occurrences} gathering all the damages ^{noted} from all the leaves, but it must be ^{noticed} that *Parrotia* fossil leaves from Willershausen are the most representative in terms of the size variability for this specific damage (Figure 3). This is certainly ^{a consequence} because of the large quantity of specimens analysed. Nevertheless, we split and compared the average of damage sizes per locality (then per *Parrotia* species) (Figure 3).

The damage trace is a curved ^{skeletonization} mine subdivided in a row of several holes. This long, curved chain of small holes usually is less than 1 cm long. However, some of the specimens can reach 1.5 cm in length, but this is quite rare. Individual holes are commonly rectangular ^{in shape} with rounded corners. The length of each hole never exceeds ^{superscript} more than 1 mm and the width of each hole, which also means ^{i.e.} the width of the mine damage, is around 0.6 mm. There is no variation of the width along the ^{course of the} damage. On average, the surface area of the damage is 4.1 mm² (± 1.4) with a length of 5.4 mm (± 1.3) and a global width of 0.6 mm (± 0.08). The number of holes can vary from 3 ^{to 12} to 12 ^{to 12}, but in most cases it is 5-8.

The small lines that separate individual holes from each other are very thin, ^{commonly} sometimes inconspicuous or missing. Although these lines are ^{indistinct} missing it is possible to infer their existence by carefully observing the internal borders of the damage. In some cases, the small lines are missing along longer portions of the mine (Figure 2, C2). The margin of the damage is marked by ^{dark} black edges. This black scar is a typical reaction from the leaf after being attacked by insect feeding and makes it possible to distinguish a herbivory trace made

A range should not be made up of two ranges. Just say 3-12.

by an insect during the leaf lifespan from a detritivorous trace made after the leaf lifespan (26,77). Overall, the path of the damage is not affected by the leaf venation. However, we noticed that the damage usually follows the secondary vein instead of removing it. Exceptionally, we noticed that damage crosses over the primary vein (Figure 2, D4) but without removing the vein. There is usually more than one damage per leaf blade; it is quite rare to observe only one damage per leaf. We also observed some cases with the entire leaf blade being covered by this damage (Figure 2, A3).

4.2 Host Plant

Based on the fossil record, there is little doubt that DT297 is exclusively found on *Parrotia* species. Investigations of fossil leaves from Willershausen were based on around 8,000 fossil specimens, comprising a large diversity of plant species more than 130 plant species/morphotypes. Macrofossils from the Shanwang collection represent a total of around 1,300 specimens and include more than 100 leaf morphotypes.

4.3 Classification

The classification of this new damage type follows the rules and terminology of the Guide to Insect (and Other) Damage Types on Compressed Plant Fossils (26). This damage is now labelled as follows: DT297.

So it is not a new damage type! - just a new designation

4.4 The specimen reference of DT297

This damage type was originally described as a trace fossil *Phagophytichnus catellarius* ichnosp. nov. by Straus (52). The fossil specimen of *P. pristina* from which *Phagophytichnus catellarius* was described belongs to the Willershausen fossil collection from Göttingen. We choose the sample GZG.W n°10626 as the main reference for DT297 (Supplement 1). This specimen is located in the Willershausen plant macrofossil collection at Göttingen University, Germany.

I trust that this is the same as the holotype of *P. catellarius*.

5. Discussion

First and foremost, it is important to mention that between the two extant *Parrotia* species and also both fossil and modern leaves, there is no difference in terms of leaf thickness, estimated by LMA (Supplement 2) based on the method from Royer et al. (2007) (79). LMA can be correlated with climatic factors (80-82), leaf nutrient availability (83,84) and furthermore, LMA can affect herbivory patterns that we can observe on the leaf blade (78,85).

5.1 DT297: a new specialist insect damage trace

Our observations demonstrate that modern *Parrotia persica* and *Parrotia subaequalis* bear the same insect feeding trace (DT297). Morphological descriptions are uniform and also statistical assessments support this observation. Indeed, measurements of the surface area, length and the width of this damage type between the *Parrotia* species and between fossil and modern specimens do not reveal any significant differences ($\alpha=0.1$) (Figure 3). The statistics may be quite weak due to the low number of measurements on modern leaves. However, as the statistics did not demonstrate any significant variations of measurements, our results indicate low size variation of the damage. (Fig. 4)

The new damage type DT297 will be considered for the next version of the "Guide to Insect (and Other) Damage Types on Compressed Plant Fossils" (26) and will be classified into the "skeletonization" functional feeding group. Figure 4 shows DT297 on *P. persica* and *P. subaequalis* leaves.

This herbivory trace is exclusively found on *Parrotia* for at least 15 Ma. Indeed, it is important to note that the outcrops mentioned in this study are not the only ones, which have been investigated for the present study. Several fossil localities in Eurasia of Cenozoic age have also been investigated and did not reveal any trace of this new damage type (20,23,29,76,86-89), neither on *Parrotia* leaves nor on any other plant species. These also include the fossil leaves of *Parrotia* from the localities Berga and Bernasso presented in Adroit et al., 2018a (27). Further, the method of identification of damage types in the leaf fossil record (26) is now used for more than ten years in numerous studies throughout the world and through the geological time periods (18,21,24,25,90-92) and none of them mentioned feeding traces such as the one presented in this study. Hence, DT297 can be considered a highly specialised skeletonization with a host specificity index of 3 (HS=3).

5.2 Specialist herbivory pattern for 15 Ma in Eurasia

Ma refers to million years ago. Here you mean R. Soc. open sci.

Myrs - because you refer to a span of time.

No. This type of text goes in the main body of the text, <https://mc.manuscriptcentral.com/rsos> figure caption - not in

These are "connecting" words, so you would not use these at the start of a paragraph.

DT297 provides direct evidence of the continuous relationship between a plant and a herbivore. So far, this is the most ancient herbivory trace specifically identified and still distinctive in the modern flora on the same plant genus. This specific damage has never changed in terms of plant host association and morphological characteristics (shape, size). At the same time, it has been distributed from western Europe to eastern Asia at least during 15 million years; a long period of time and a large geographic area which is characterized by marked environmental differences.

Indeed, the warm climate during the Middle Miocene Climatic Optimum (17 Ma - 15 Ma) (93), followed by progressive cooling during the Middle Miocene Climate Transition (15 Ma - 13 Ma) (94), and the onset of glacial-interglacial cycles from the Middle Pleistocene Transition (1.2 Ma - 0.7 Ma) onwards (95) occurred between the first known traces of DT297 and now. In addition, orogenesis was extremely important in Eurasia, especially with the rise of the Tibet-Qinghai Plateau (96-98), which formed a barrier between eastern Asia (*P. subaequalis*) and the Caucasus - Europe (*P. persica*).

Thus, the damage DT297 represents remarkable stasis of a feeding trace. Numerous studies have demonstrated shifts of herbivory during different geological period events (16,23,99,100). Those herbivory changes are mostly due to climate variations because of the insect physiology (101-105) and, in some cases, due to interruptions of gene exchange between plant and insect species (106,107) created by the emergence of new landforms.

Although the specific damage on *Parrotia* during (at least) 15 Ma can be utilised to reconstruct the trophic structure of *Parrotia* in its environment, it is very difficult to determine the insect causing this damage. In his manuscript from 1977, Straus suggested that this trace fossil could have been produced by Chrysomelidae larvae. Based on our comparisons with known insect feeding from the literature we suggest that DT297 could have been caused by insects belonging to subfamily Galerucinae/Alticinae, probably by *Altica* which was widely distributed in Eurasia from at least the Eocene (108-110) and is today known in the Hyrcanian forest (111) and the Yangtze River valley (112). Both subfamilies are recorded in the Yangtze River valley and the Hyrcanian forest region with some endemic species of these regions (113-116).

The present specific damage shared between *P. subaequalis* and *P. persica* today completely isolated from each other, is a direct evidence that they occupied a common ecological niche, which is today separated into two geographic areas, being an example of vicariance biogeography (117). In the literature, the congenicity of the *Parrotia* species between eastern Asia and Caucasus is still not clear and even led to consider the Asian Hamamelidaceae as *Shaniodendron subaequalis* (= *P. subaequalis*) for some researchers (40,65). Based on molecular point of view, Li et al. (1999) used internal transcribed spacers from nuclear gene, into the Hamamelidaceae and supported that *Parrotia* and *Shaniodendron* are a monophyletic group (118,119). Additional studies based on the chloroplast gene "matK" organised *Parrotia* and *Shaniodendron* as two distinct taxa (120,121). A recent study even describes the whole chloroplast genome of the *P. subaequalis* (122). However, chloroplast markers usually do not reconstruct taxonomic but biogeographic relationships (123-127). Along this line, this new DT297 can actually contribute to better understand some of the shared biogeographic history of the two host lineages in western Eurasia and eastern Asia and support the accordance between these *Parrotia* species presented in this study.

From another perspective, in a continuous co-evolution between insect attack strategies and plant defence strategies (128,129), it is difficult to explain why such a specific herbivory trace has never changed during 15 Ma. One hypothesis is that this insect damage can be mutualistic in some cases. Although this is debated within the scientific community, Agrawal (2000) (130) demonstrated, based on the plant fitness, that certain types of insect feeding could be mutualistic interactions between the insect and the plant. Moreover, a recent meta-analysis of hundreds of scientific publications (131) about the "overcompensation" for insect herbivory also supports this hypothesis. However, to focus only on plant fitness is not enough to discuss mutualism as a whole, because mutualism implies also an evolutionary history of the plant-animal interaction (132), in which a specific feeding trace, such as DT297, can be considered as a direct evidence.

5.3 DT297 more common in the fossil record than on modern leaves

We observed many more examples of DT297 on fossil than on modern leaves. The most striking difference is seen in *P. persica* and its fossil analogue *P. pristina*, as we observed around 500 fossil specimens from Willershausen (27) versus more than 2,300 modern specimens in its modern range in the Hyrcanian forest (133) and yet the large majority of the DT297 has been observed on the fossil leaves (Supplement 3). This is also true for the Chinese fossil leaf assemblages, which recorded almost ten times more occurrences of the DT297 than the modern *P. subaequalis* leaves from the Yixing forest area (Supplement 3). This unexpected pattern can be explained in various ways.

These significant differences of occurrences could indicate an ecological change for this specialist plant-insect interaction. The populations of insects specialised on *Parrotia* could have significantly decreased during

the last 15 ^{yrs} Ma until they also became relictual in the Hyrcanian and Yixing forests. The large climatic changes during the Miocene (93,134,135) and introduction of glacial-interglacial cycles in the Quaternary (53,95) had a huge impact on numerous plant species population and their distribution ^{area} (56,136–138), including *Parrotia* (139–141). However, there are some examples of insect species ^{that} which survived the glacial-interglacial cycles and recolonized the same area such as ^{for example} the arctic-alpine insect species *Arcynopteryx dichroa* ^{the} in Central European highlands (142).

Sampling bias ^{es} could also have caused the marked differences of occurrences between fossil and modern leaves. The modern leaves sampled ^{actually} represent only one or two years of leaf shedding, ^{whereas} while fossil leaves may represent many years of leaf production and hence environmental variation such as dry or wet years. Fossil leaves from Shanwang ^{was recovered from several} belong to different layers (19 sub-units in total) of diatomaceous sedimentary rocks (143) and represent a maximum of 3 million years of elapsed time (32,67,70). ^{At} Willershausen, fossils were collected from a clay pit and fossils from different layers were mixed ^{based on lithological differences and contrasting} however due to the rock and differences of fossil preservation, it is certain that the leaves from this outcrop represent many years of deposition (144,145). Plant-insect interactions can significantly change from one year to another due to a multitude of factors such as climate ^{variability} seasonality (146–149). Thus, insect feeding observed in the fossil record is generally more representative of the global herbivory pattern on *Parrotia* taxa than the observations made on modern leaf litter.

Only increased sampling efforts ^{for} of *Parrotia* leaves from modern sites and fossil ^{assemblages enable} outcrops will allow to better characterize ^{ation of the} this difference of DT occurrences. Accordingly, one of the main objectives of ^{our} this present ^{a study} publication was to thoroughly describe the new damage type DT297 in order to provide a basis for more comprehensive investigations in the future.

6. Conclusion

This study highlights and describes a new long-term ^{mode of} herbivory, expressed as a curved ^{represented} chain of holes, which is exclusively observable ^{on the} *Parrotia* plant species. Morphological description and measurements confirm that the ^{new} damage type (DT297) is exactly the same on those two species. In addition, it appears that this relationship between herbivore and *Parrotia* plant species has ^{persisted} been persisting for at least 15 Ma. This new damage type ^{provides} is a direct evidence, quite rare in palaeoecology, of long-term relationship between a plant species and its herbivore. ^{currently} Actually, DT297 is ^{for now} the most specific long-term herbivory trace ^{still} identifiable on ^{the same plant lineage today}. Henceforth, in order to better understand this interaction, fieldwork should be made in the Hyrcanian and/or Yixing forests in order to directly observe the insect species causing ^{this} the distinct damage ^{form} DT297.

The discovery ^{of} the continuous presence of this damage type ^{over 15 Mys} demonstrates a robust and long-term plant-herbivore association, and provides ^{improved} new evidence for a shared biogeographic history of the two host plants. This may have implications for a ^{better} understanding of phylogenetic relationships between the western Eurasian and East Asian host plant species.

Acknowledgments

We would like to thank Dr. Thomas Denk ^{for sharing} to share some *Parrotia* fossil specimens stored in Naturhistoriska riksmuseet of Stockholm ^{from the Willershausen outcrop} and for his considerable support to improve our manuscript. Thank you also to Dr. Thomas Bastelberger ^{for sharing} with us some *P. pristina* samples ^{from} of his private collection of ^{fossils from} Willershausen. Thanks are due to Prof Anurag Agrawal (Cornell University) for discussions about the topic of mutualism ^{and} herbivory. ^{we} Also thanks to Dr. Vincent Girard and Dr. Conrad Labandeira for fruitful discussions, which helped improving our manuscript. Angela Nekes (Siegen University, Germany) helped with the German translation of an old manuscript about the Willershausen outcrop.

Funding Statement

We did not receive any specific grants for this project but enjoyed support from the Deutsche Forschungsgemeinschaft (DFG) to visit Willershausen fossil collection in some museums of Germany ^{the} and Strategic Priority Research Program of the Chinese Academy of Sciences (XDB26000000) and National Natural Science Foundation of China (41688103) to work on this publication.

Data Accessibility

All data are accessible in the supplementary files of this manuscript.

Competing Interests

We have no competing interests.

Authors' Contributions

B.A., X.Z., T.W. collect fossil and/or modern leaves; B.W. obtained the funding; B.A. made the analysis; B.A. B.W. wrote the draft; B.W., J.T., T.W. contributed to the discussion
All authors approved the publication.

References

1. Brooker RW, Callaghan TV. The Balance between Positive and Negative Plant Interactions and Its Relationship to Environmental Gradients: A Model. *Oikos*. 1998;81(1):196-207.
2. May RM. Stability in ecosystems: some comments. In: van Dobben WH, Lowe-McConnell RH, éditeurs. *Unifying Concepts in Ecology: Report of the plenary sessions of the First international congress of ecology, The Hague, the Netherlands, September 8–14, 1974* [Internet]. Dordrecht: Springer Netherlands; 1975 [cité 14 oct 2019]. p. 161-8. Disponible sur: https://doi.org/10.1007/978-94-010-1954-5_13
3. Tschirhart J. General Equilibrium of an Ecosystem. *Journal of Theoretical Biology*. 7 mars 2000;203(1):13-32.
4. Heller NE, Zavaleta ES. Biodiversity management in the face of climate change: A review of 22 years of recommendations. *Biological Conservation*. janv 2009;142(1):14-32.
5. IPCC, éditeur. *Climate change 2007: the physical science basis: contribution of Working Group I to the Fourth Assessment Report of the Intergovernmental Panel on Climate Change*. Cambridge; New York: Cambridge University Press; 2007. 996 p.
6. Turner BLI, Clark W C, Kates R W, Richards J F, Mathews J T, Meyer W B. *The Earth was formed by Human Action: Global and Regional Changes in the Past 300 Years*. Cambridge Univ Press, Cambridge. 1990;
7. Hillstrom ML, Lindroth RL. Elevated atmospheric carbon dioxide and ozone alter forest insect abundance and community composition: Carbon dioxide and ozone alter forest insect communities. *Insect Conservation and Diversity*. 24 oct 2008;1(4):233-41.
8. Goulson D, Nicholls E, Botías C, Rotheray EL. Bee declines driven by combined stress from parasites, pesticides, and lack of flowers. *Science*. 27 mars 2015;347(6229):1255-957.
9. Coley PD, Barone JA. Herbivory and Plant Defenses in Tropical Forests. *Annual Review of Ecology and Systematics*. 1996;27(1):305-35.
10. Grimaldi D. The Co-Radiations of Pollinating Insects and Angiosperms in the Cretaceous. *Annals of the Missouri Botanical Garden*. 1999;86(2):373-406.
11. McGhee GR. *Convergent Evolution: Limited Forms Most Beautiful*. MIT Press; 2011. 335 p.
12. Forister ML, Novotny V, Panorska AK, Baje L, Basset Y, Butterill PT, et al. The global distribution of diet breadth in insect herbivores. *Proceedings of the National Academy of Sciences*. 13 janv 2015;112(2):442-7.
13. Lewinsohn TM, Novotny V, Basset Y. Insects on Plants: Diversity of Herbivore Assemblages Revisited. *Annual Review of Ecology, Evolution, and Systematics*. 2005;36(1):597-620.
14. Nyman T. To speciate, or not to speciate? Resource heterogeneity, the subjectivity of similarity, and the macroevolutionary consequences of niche-width shifts in plant-feeding insects. *Biological Reviews*. 2010;85(2):393-411.
15. Coley PD, Aide TM. Comparisons of herbivory and plant defenses in temperate and tropical broad-leaved forests. In: *Plant-Animal Interactions: Evolutionary Ecology in Tropical and Temperate Regions*. Wiley-Interscience. New York, USA: Peter W. Price, Thomas M. Lewinsohn, G. Wilson Fernandes, Woodruff W. Benson; 1991. p. 25-49.
16. Currano ED, Labandeira CC, Wilf P. Fossil insect folivory tracks paleotemperature for six million years. *Ecological Monographs*. 2010;80(4):547–567.
17. Zvereva EL, Kozlov MV. Consequences of

Do you need issue numbers?
Hyphen or endash?
Do you really need the date?

- simultaneous elevation of carbon dioxide and temperature for plant-herbivore interactions: a metaanalysis. *Global Change Biology* (Janv) 2006;12(1):27-41.
18. Currano ED, Labandeira CC, Wilf P. Dynamics of plant-insect interactions during late Paleocene and early Eocene environmental perturbations in the Bighorn Basin, Wyoming, USA. *Climatic and Biotic Events of the Paleogene*. 2009;44-6.
19. Labandeira C. The four phases of plant-arthropod associations in deep time. *Geologica Acta*. 2006;4(4):409.
20. Labandeira CC, Kustatscher E, Wappler T. *Floral Assemblages and Patterns of Insect Herbivory during the Permian to Triassic of Northeastern Italy*. Wong WO, éditeur. *PLoS ONE*. 9 nov 2016;11(11):e0165205.
21. Labandeira CC, Dilcher DL, Davis DR, Wagner DL. Ninety-seven million years of angiosperm-insect association: paleobiological insights into the meaning of coevolution. *PNAS*. 1994;91(25):12278-82.
22. Wappler T. Insect herbivory close to the Oligocene-Miocene transition — A quantitative analysis. *Palaeogeography, Palaeoclimatology, Palaeoecology*. juin 2010;292(3-4):540-50.
23. Wappler T, Currano ED, Wilf P, Rust J, Labandeira CC. No post-Cretaceous ecosystem depression in European forests? Rich insect-feeding damage on diverse middle Palaeocene plants, Menat, France. *Proceedings of the Royal Society B. Biological Sciences*. 22 déc 2009;276(1677):4271-7.
24. Wilf P. Insect-damaged fossil leaves record food web response to ancient climate change and extinction. *New Phytologist*. mai 2008;178(3):486-502.
25. Wilf P, Labandeira CC, Johnson KR, Coley PD, Cutter AD. Insect herbivory, plant defense, and early Cenozoic climate change. *Proceedings of the National Academy of Sciences*. 2001;98(11):6221-6226.
26. Labandeira CC, Wilf P, Johnson KR, Marsh F. Guide to insect (and other) damage types on compressed plant fossils. *Smithsonian Institution, National Museum of Natural History, Department of Paleobiology, Washington, DC* [Internet]. 2007 [cité 27 avr 2014]; Disponible sur: <http://citeseerx.ist.psu.edu/viewdoc/download?doi=10.1.1.2.12.4539&rep=rep1&type=pdf>
27. Adroit B, Girard V, Kunzmann L, Terral J-F, Wappler T. Plant-insect interactions patterns in three European paleoforests of the late-Neogene-early-Quaternary. *PeerJ*. 2018;6(5075):24.
28. Knobloch E. Der pliozäne laubwald Laubwald von willershausen Willershausen am harz Harz (Mitteleuropa). *Documenta naturae*. 1998;120:1-302.
29. Adroit B, Malekhosseini M, Girard V, Abedi M, Rajaei H, Terral J-F, et al. Changes in pattern of plant-insect interactions on the Persian ironwood (*Parrotia persica*, Hamamelidaceae) over the last 3 million years. *Review of Palaeobotany and Palynology*. nov 2018;258:22-35.
30. Parsa A. *Flore de l'Iran*. Vol. 2. Iran: Ministère de l'éducation. Museum d'histoire naturelle de Tehran; 1943. 613 p.
31. Talebi KS, Sajedi T, Pourhashemi M. *Forests of Iran* [Internet]. Dordrecht: Springer Netherlands; 2014 [cité 11 mars 2015]. (Plant and Vegetation; vol. 10). Disponible sur: <http://link.springer.com/10.1007/978-94-007-7371-4>
32. Yang H, Yang S. The Shanwang fossil biota in eastern China: a Miocene Konservat-Lagerstätte in lacustrine deposits. *Lethaia*. 1 déc 1994;27(4):345-54.
33. Zhang J-F, Sun B, Zhang X -y. *Sedimentary geology of Shanwang Basin*. Beijing: Science Press; 1994. 200 p.
34. Hao RM, Wei HT. A new combination of Hamamelidaceae. *Acta Phytotaxonomica Sinica* 36. 1998;36(80).
35. Mai DH. *Tertiäre Vegetationsgeschichte Europas: Methoden und Ergebnisse* [Internet]. Stuttgart, Germany: Fischer; 1995. Disponible sur: <https://books.google.de/books?id=wTvwAAAAMAAJ>
36. Zhilin SG. History of the development of the temperate forest flora in Kazakhstan, U.S.S.R. from the Oligocene to the Early Miocene. *Bot Rev*. oct 1989;55(4):205-330.

lower case?

italics

37. Tanai T. On the Hamamelidaceae from the Paleogene of Hokkaido, Japan. *Transactions and Proceedings of the Palaeontological Society of Japan*. 1967;(66):56-62.
38. Pavlyutkin BI, Yabe A, Golozoubov VV, Simanenko LF. Miocene Floral Changes in the Circum-Japan Sea Areas—Their Implications in the Climatic Changes and the Time of Japan Sea Opening. *Mem Natl Mus Nat Sci*, Tokyo. 2016;(51):109-23.
39. Hu HH, Chaney RW. A Miocene flora from Shantung province, China. *Carnegie Institute of Washington Publication*. 1940;(507):1-147.
40. Wang X-Q, Li H-M. Discovery of another living fossil - *Shaniodendron subaequale* (H.T.Chang) Deng et al. in China - Clearing up paleobotanists' a long-term doubt. *Acta Palaeontologica Sinica*. 2000;39(Sup.):308-17.
41. Wu ZY, Raven PH, Hong DY. *Flora of China*. Vol. 9 (Pittosporaceae through Connaraceae). Science Press, Beijing, and Missouri Botanical Garden Press, St. Louis. 2003;
42. Coombes AJ, Debreczy Z. *The Book of Leaves* [Internet]. United Kingdom: Ivy Press; 2015. Disponible sur: <https://books.google.de/books?id=UvENswEACAAJ>
43. Leroy SA, Roiron P. Latest Pliocene pollen and leaf floras from Bernasso palaeolake (Escandorgue Massif, Hérault, France). *Review of Palaeobotany and Palynology*. 1996;94(3):295–328.
44. Yosefzadeh H, Tabari M, Akbarinia M, Akbarian MR, Bussotti F. Morphological plasticity of *Parrotia persica* leaves in eastern Hyrcanian forests (Iran) is related to altitude. *Nordic Journal of Botany*. 16 avr 2010;28(3):344-9.
45. Bachmann GH, Ehling BC, Eichner R, Schwab M. *Geologie von Sachsen-Anhalt*. Schweizerbart Science Publishers. Stuttgart, Germany; 2008. 689 p.
46. Boulter MC, N.L.B. Hubbard R, Kvaček Z. A comparison of intuitive and objective interpretations of Miocene plant assemblages from north Bohemia. *Palaeogeography, Palaeoclimatology, Palaeoecology*. mars 1993;101(1-2):81-96.
47. Buzek C. Tertiary flora of the Northern Part of the Petipsy Area (North-Bohemian Basin) [Internet]. Stuttgart, Germany: Schweizerbart Science Publishers; 1971. Disponible sur: http://www.schweizerbart.de/publications/detail/isbn/9783510991037/Rozpravy_Buzek_Tertiary_Flora_Bd_36
48. Denk T, Güner TH, Kvaček Z, Bouchal JM. The early Miocene flora of GÜvem (Central Anatolia, Turkey): a window into early Neogene vegetation and environments in the Eastern Mediterranean. *Acta Palaeobotanica*. 1 déc 2017;57(2):237-338.
49. Ferguson DK, Knobloch E. A fresh look at the rich assemblage from the Pliocene sink-hole of Willershausen, Germany. *Review of Palaeobotany and Palynology*. 1998;101(1):271–286.
50. Macaluso L, Martinetto E, Vigna B, Bertini A, Cilia A, Teodoridis V, et al. Palaeofloral and stratigraphic context of a new fossil forest from the Pliocene of NW Italy. *Review of Palaeobotany and Palynology*. janv 2018;248:15-33.
51. Martinetto E. Leaves of terrestrial plants from the Pliocene shallow marine and transitional deposits of Asti (Piedmont, NW Italy). *Bollettino della Società Paleontologica Italiana*. 2003;42(1-2):75-11.
52. Straus A. Gallen, Minen und andere Fraßspuren im Pliozän von Willershausen am Harz. *Verhandlungen des Botanischen Vereins der Provinz Brandenburg*. 1977;113:41-80.
53. Hilgen FJ. Astronomical calibration of Gauss to Matuyama sapropels in the Mediterranean and implication for the geomagnetic polarity timescale. *Earth Planet Sci Lett*. 1991;104:226-44.
54. Suc J-P. Analyse pollinique de dépôts Pliocène du sud du massif basaltique de l'Escandorgue (Site de Bernasso, Lunas, Hérault, France). *Pollen et Spores*. 1978;20(4):497-512.
55. Akhiani H, Ziegler H. Photosynthetic pathways and habitats of grasses in Golestan National Park (NE Iran), with an emphasis on the C4-grass dominated rock communities. *Phytocoenologia*. 2002;32(3):455–501.
56. Arpe K, Leroy SAG. The Caspian Sea Level forced by the atmospheric circulation, as observed and modelled. *Quaternary International*. oct

Some abbreviational but others not

- 2007;173-174:144-52.
57. Li H, Yue C, Zhang Y, Shao S, Yu L. Advance of Research on *Parrotia subaequalis* (in Chinese). *Journal of Zhejiang Forestry Science and Technology*. 2012;32(5):79-84.
58. Li J-H, Bogle AL, Klein AS. Close relationship between *Shaniodendron* and *Parrotia* (Hamamelidaceae), evidence from its sequences of nuclear ribosomal DNA. *Acta Phytotaxonomica Sinica*. 1997;(35):481-3.
59. Fang Y, Jin Y, Deng M, Yang Q, Li B. Macroscopic and Microscopic Structure of *Parrotia subaequalis* Leaves and its systemic significance (In Chinese). *Plant Resources and Environment*. 1997;36-42.
60. Geng Q, Yao Z, Yang J, He J, Wang D, Wang Z, et al. Effect of Yangtze River on population genetic structure of the relict plant *Parrotia subaequalis* in eastern China. *Ecology and Evolution*. oct 2015;5(20):4617-27.
61. Yang Q. Preliminary report on the study of *Shaniodendron subaequalis* (in Chinese). *Journal of Jiangsu Forestry Science and Technology*. 1994;(1):14-8.
62. Zhou Z, Momohara A. Fossil History of Some Endemic Seed Plants of East Asia and Its Phytogeographical Significance (in Chinese). *Journal of Plant Classification and Resources*. 2005;(27):449-70.
63. Manchester SR, Chen Z, Geng B, Tao J. Middle Eocene flora of Huadian, Jilin Province, Northeastern China. *Acta Palaeobotanica*. 2005;45(1):3-26.
64. Yao Z, Wang Z, Yan C, Dong Z, Xu W, Wei N, et al. The photosynthesis response to different light intensity for the endangered plant *Parrotia subaequalis* (in Chinese). *Journal of Nanjing Forestry University*. 2010;34(3):83-8.
65. Deng MB, Wei HT, Wang XQ. *Shaniodendron subaequalis* (H.T. Chang) M.B. Deng, H.T. Wei & X.Q. Wang. *Acta Phytotaxonomica Sinica*. 1992;30(1):59-61.
66. Gradstein F, Ogg JG, Smith AG, Bleeker W, Lourens LJ. A new Geologic Time Scale, with special reference to Precambrian and Neogene. *Episodes*. 2004;27(2):83-100.
67. He H, Deng C, Pan Y, Deng T, Luo Z, Sun J, et al. New ⁴⁰Ar/³⁹Ar dating results from the Shanwang Basin, eastern China: Constraints on the age of the Shanwang Formation and associated biota. *Physics of the Earth and Planetary Interiors*. juill 2011;187(1-2):66-75.
68. Liang M-M, Bruch A, Collinson M, Mosbrugger V, Li C-S, Sun Q-G, et al. Testing the climatic estimates from different palaeobotanical methods: an example from the Middle Miocene Shanwang flora of China. *Palaeogeography, Palaeoclimatology, Palaeoecology*. oct 2003;198(3-4):279-301.
69. Qiu Z. The Chinese Neogene Mammalian Biochronology — Its Correlation with the European Neogene Mammalian Zonation. In: Lindsay EH, Fahlbusch V, Mein P, éditeurs. *European Neogene Mammal Chronology* [Internet]. Boston, MA: Springer US; 1989 [cité 24 sept 2019]. p. 527-56. (NATO ASI Series). Disponible sur: https://doi.org/10.1007/978-1-4899-2513-8_32
70. Sun Q, Collinson ME, Li C-S, Wang Y, Beerling DJ. Quantitative reconstruction of palaeoclimate from the Middle Miocene Shanwang flora, eastern China. *Palaeogeography, Palaeoclimatology, Palaeoecology*. 2002;(180):315-29.
71. Liu G, Leopold EB. Paleocology of a Miocene flora from the Shanwang formation, Shandong province, northern East China. *Palynology*. déc 1992;16(1):187-212.
72. Zhao J. Qinling East Extension and Jiangsu Natural Environment (in Chinese). *Jiangsu Geology*. 2008;(79).
73. Cornelissen T, Stiling P. Sex-biased herbivory: a meta-analysis of the effects of gender on plant-herbivore interactions. *Oikos*. 2005;111(3):488-500.
74. Leckey EH, Smith DM, Nufio CR, Fornash KF. Oak-insect herbivore interactions along a temperature and precipitation gradient. *Acta Oecologica*. nov 2014;61:1-8.
75. Reynolds BC, Crossley DA Jr. Spatial Variation in Herbivory by Forest Canopy Arthropods Along an Elevation Gradient. *Environmental Entomology*. 1997;26(6):1232-9.

Why
 Doi
 for
 some
 but not
 for
 others?

lower
 case
 ↑

76. Su T, Adams JM, Wappler T, Yong-Jiang H, Jacques FMB, Liu Y-S (Christopher), et al. Resilience of plant-insect interactions in an oak lineage through Quaternary climate change. *Paleobiology*. 2015;41:174-86.
77. Labandeira CC. The history of associations between plants and animals. *Plant–Animal Interactions: An Evolutionary Approach*. 2002;248:261.
78. de la Riva EG, Olmo M, Poorter H, Uberta JL, Villar R. Leaf Mass per Area (LMA) and Its Relationship with Leaf Structure and Anatomy in 34 Mediterranean Woody Species along a Water Availability Gradient. *Armas C, éditeur. PLoS ONE*. 11 févr 2016;11(2):e0148788.
79. Royer DL, Sack L, Wilf P, Lusk CH, Jordan GJ, Niinemets U, et al. Fossil leaf economics quantified: calibration, Eocene case study, and implications. *Paleobiology*. 2007;33(4):574-89.
80. Givnish T. Adaptive significance of evergreen vs. deciduous leaves: solving the triple paradox. *Silva Fenn* [Internet]. 2002 [cité 17 oct 2019];36(3). Disponible sur: <http://www.silvafennica.fi/article/535>
81. Ishida A, Diloksumpun S, Ladpala P, Staporn D, Panuthai S, Gamo M, et al. Contrasting seasonal leaf habits of canopy trees between tropical dry-deciduous and evergreen forests in Thailand. *Tree Physiol*. 1 mai 2006;26(5):643-56.
82. Prior LD, Bowman DMJS, Eamus D. Seasonal differences in leaf attributes in Australian tropical tree species: family and habitat comparisons. *Functional Ecology*. 2004;18(5):707-18.
83. Fonseca CR, Overton JM, Collins B, Westoby M. Shifts in trait-combinations along rainfall and phosphorus gradients. *Journal of Ecology*. 2000;88(6):964-77.
84. McDonald PG, Fonseca CR, Overton JM, Westoby M. Leaf-size divergence along rainfall and soil-nutrient gradients: is the method of size reduction common among clades? *Functional Ecology*. 2003;17(1):50-7.
85. Currano ED, Wilf P, Wing SL, Labandeira CC, Lovelock EC, Royer DL. Sharply increased insect herbivory during the Paleocene–Eocene Thermal Maximum. *Proceedings of the National Academy of Sciences*. 2008;105(6):1960–1964.
86. Gunkel S, Wappler T. Plant-insect interactions in the upper Oligocene of Enspel (Westerwald, Germany), including an extended mathematical framework for rarefaction. *Palaeobiodiversity and Palaeoenvironments*. mars 2015;95(1):55-75.
87. Knor S, Prokop J, Kvaček Z, Janovský Z, Wappler T. Plant–arthropod associations from the Early Miocene of the Most Basin in North Bohemia—Palaeoecological and palaeoclimatological implications. *Palaeogeography, Palaeoclimatology, Palaeoecology*. mars 2012;321-322:102-12.
88. Kunzmann L, Morawek K, Müller C, Schröder I, Wappler T, Grein M, et al. A Paleogene leaf flora (Profen, Sachsen-Anhalt, Germany) and its potentials for palaeoecological and palaeoclimate reconstructions. *Flora* [Internet]. nov 2018 [cité 6 févr 2019]; Disponible sur: <https://linkinghub.elsevier.com/retrieve/pii/S0367253018306431>
89. Wappler T, Grímsson F. Before the ‘Big Chill’: Patterns of plant-insect associations from the Neogene of Iceland. *Global and Planetary Change*. juill 2016;142:73-86.
90. Dong J, Sun B, Mao T, Yan D, Liu C, Wang Z, et al. Liquidambar (Altingiaceae) and associated insect herbivory from the Miocene of southeastern China. *Palaeogeography, Palaeoclimatology, Palaeoecology*. mai 2018;497:11-24.
91. Ma F-J, Ling C-C, Ou-Yang M-S, Yang G-M, Shen X-P, Wang Q-J. Plant–insect interactions from the Miocene (Burdigalian–Langhian) of Jiangxi, China. *Review of Palaeobotany and Palynology*. janv 2020;104:176.
92. McDonald CM, Francis JE, Compton SGA, Haywood A, Ashworth AC, Hinojosa LF, et al. Herbivory in Antarctic fossil forests: evolutionary and palaeoclimatic significance. 2007 [cité 25 sept 2017]; Disponible sur: <http://digitalcommons.unl.edu/andrillaffiliates/4/>
93. Böhme M. The Miocene Climatic Optimum: evidence from ectothermic vertebrates of Central Europe. *Palaeogeography, Palaeoclimatology,*

- 1
2
3
4
5
6
7
8
9
10
11
12
13
14
15
16
17
18
19
20
21
22
23
24
25
26
27
28
29
30
31
32
33
34
35
36
37
38
39
40
41
42
43
44
45
46
47
48
49
50
51
52
53
54
55
56
57
58
59
60
- Palaeoecology. 15 juin 2003;195(3):389-401.
94. Frigola A, Prange M, Schulz M. Boundary conditions for the Middle Miocene Climate Transition (MMCT v1.0). *Geoscientific Model Development*. 24 avr 2018;11(4):1607-26.
95. Clark PU, Archer D, Pollard D, Blum JD, Rial JA, Brovkin V, et al. The middle Pleistocene transition: characteristics, mechanisms, and implications for long-term changes in atmospheric pCO₂. *Quaternary Science Reviews*. déc 2006;25(23-24):3150-84.
96. Kutzbach JE, Prell WL, Ruddiman WmF. Sensitivity of Eurasian Climate to Surface Uplift of the Tibetan Plateau. *The Journal of Geology*. 1 mars 1993;101(2):177-90.
97. Zhisheng A, Kutzbach JE, Prell WL, Porter SC. Evolution of Asian monsoons and phased uplift of the Himalaya–Tibetan plateau since Late Miocene times. *Nature*. mai 2001;411(6833):62-6.
98. Bartholomé E, Belward AS. GLC2000: a new approach to global land cover mapping from Earth observation data. *International Journal of Remote Sensing*. mai 2005;26(9):1959-77.
99. Adams JM, Brusa A, Soyeong A, Ainuddin AN. Present-day testing of a paleoecological pattern: Is there really a latitudinal difference in leaf-feeding insect-damage diversity? *Review of Palaeobotany and Palynology*. août 2010;162(1):63-70.
100. Wilf P, Johnson KR, Cúneo NR, Smith ME, Singer BS, Gandolfo MA. Eocene plant diversity at Laguna del Hunco and Río Pichileufú, Patagonia, Argentina. *The American Naturalist*. 2005;165(6):634–650.
101. Chown SL, Gaston KJ. Body size variation in insects: a macroecological perspective. *Biological Reviews*. 2010;85(1):139-69.
102. Clapham ME, Karr JA. Environmental and biotic controls on the evolutionary history of insect body size. *Proceedings of the National Academy of Sciences*. 3 juill 2012;109(27):10927-30.
103. Harrison JF, Kaiser A, VandenBrooks JM. Atmospheric oxygen level and the evolution of insect body size. *Proceedings of the Royal Society B: Biological Sciences*. 7 juill 2010;277(1690):1937-46.
104. Schmidt-Nielsen K, Knut S-N. *Scaling: Why is Animal Size So Important?* Cambridge University Press; 1984. 260 p.
105. Siemann E, Tilman D, Haarstad J. Insect species diversity, abundance and body size relationships. *Nature*. 1996;(380):704-6.
106. Maron JL, Agrawal AA, Schemske DW. Plant–herbivore coevolution and plant speciation. *Ecology*. 20 mai 2019;e02704.
107. Stenberg JA, Witzell J, Ericson L. Tall herb herbivory resistance reflects historic exposure to leaf beetles in a boreal archipelago age-gradient. *Oecologia*. 1 juin 2006;148(3):414-25.
108. Nadein KS, Perkovsky EE. Small and common: the oldest tropical Chrysomelidae (Insecta: Coleoptera) from the lower Eocene Cambay amber of India. *Alcheringa: An Australasian Journal of Palaeontology*. 2 oct 2019;43(4):597-611.
109. Jolivet P, Petitpierre E, Hsiao TH. *Biology of Chrysomelidae*. Springer Science & Business Media; 2012. 624 p.
110. Bukejs A, Konstantinov A. New genus of flea beetle (Coleoptera: Chrysomelidae: Galerucinae: Alticini) from the Upper Eocene Baltic amber. *Insecta Mundi* [Internet]. 10 juin 2013; Disponible sur: <https://digitalcommons.unl.edu/insectamundi/811>
111. Shumilovskikh LS, Hopper K, Djamaali M, Ponel P, Demory F, Rostek F, et al. Landscape evolution and agro-sylvo-pastoral activities on the Gorgan Plain (NE Iran) in the last 6000 years. *The Holocene*. 1 oct 2016;26(10):1676-91.
112. Sun J-H, Liu Z-D, Britton KO, Cai P, Orr D, Hough-Goldstein J. Survey of phytophagous insects and foliar pathogens in China for a biocontrol perspective on kudzu, *Pueraria montana* var. *lobata* (Willd.) Maesen and S. Almeida (Fabaceae). *Biological Control*. 1 janv 2006;36(1):22-31.
113. Yanyu W. The species diversity of subfamily alticinae (Coleoptera: Chrysomelidae) communities in Wuyishan nature reserve. *Wuyi Science journal*. 2000;
114. Wu Y, Yuan D. Biodiversity and Conservation in China: A View from Entomologists. *Insect Science*.

CHECK
REF
STYLE
THROUGHOUT

- 1997;4(2):95-111.
115. Adeli E. Beitrag zur Kenntnis der im Forst schädlichen Insekten des Iran: I. Coleoptera. Zeitschrift für Angewandte Entomologie. 26 août 2009;70(1-4):8-14.
116. Wang SY. Coleoptera, Chrysomelidae: Chrysomelinae. In: Insects of the three Gorge Reservior Area of Yangtze River. X. K. Yang. Chongqing Publishing House, Chongqing; 1997. p. 855-62.
117. Wiley EO. Vicariance Biogeography. Annual Review of Ecology and Systematics. 1988;19(1):513-42.
118. Li J, Bogle AL, Klein AS. Phylogenetic relationships of the Hamamelidaceae inferred from sequences of internal transcribed spacers (ITS) of nuclear ribosomal DNA. Am J Bot. juill 1999;86(7):1027-37.
119. Li J, Bogle AL, Klein AS. Phylogenetic relationships in the Hamamelidaceae: Evidence from the nucleotide sequences of the plastid genematK. Pl Syst Evol. sept 1999;218(3-4):205-19.
120. Li J, Bogle L, Donoghue MJ. Phylogenetic relationships in the Hamamelidoideae inferred from sequences of trn non-coding regions of chloroplast DNA. Harvard Papers in Botany. 1999;4(1):343-56.
121. Xiang X, Xiang K, Ortiz RDC, Jabbour F, Wang W. Integrating palaeontological and molecular data uncovers multiple ancient and recent dispersals in the pantropical Hamamelidaceae. J Biogeogr. nov 2019;46(11):2622-31.
122. Zhang Y-Y, Shi E, Yang Z-P, Geng Q-F, Qiu Y-X, Wang Z-S. Development and Application of Genomic Resources in an Endangered Palaeoendemic Tree, Parrotia subaequalis (Hamamelidaceae) From Eastern China. Front Plant Sci. 1 mars 2018;9:246.
123. Mengoni A, Gonnelli C, Brocchini E, Galardi F, Pucci S, Gabbriellini R, et al. Chloroplast Genetic Diversity and Biogeography in the Serpentine Endemic Ni-Hyperaccumulator Alyssum bertolonii. The New Phytologist. 2003;157(2):349-56.
124. Mummenhoff K, Brüggemann H, Bowman JL. Chloroplast DNA phylogeny and biogeography of Lepidium (Brassicaceae). American Journal of Botany. 2001;88(11):2051-63.
125. Der JP, Thomson JA, Stratford JK, Wolf PG. Global chloroplast phylogeny and biogeography of bracken (Pteridium; Dennstaedtiaceae). American Journal of Botany. 2009;96(5):1041-9.
126. Dong W, Liu J, Yu J, Wang L, Zhou S. Highly Variable Chloroplast Markers for Evaluating Plant Phylogeny at Low Taxonomic Levels and for DNA Barcoding. PLoS One [Internet]. 12 avr 2012 [cité 22 avr 2020];7(4). Disponible sur: <https://www.ncbi.nlm.nih.gov/pmc/articles/PMC3325284/>
127. Yan M, Xiong Y, Liu R, Deng M, Song J. The Application and Limitation of Universal Chloroplast Markers in Discriminating East Asian Evergreen Oaks. Front Plant Sci [Internet]. 8 mai 2018 [cité 22 avr 2020];9. Disponible sur: <https://www.ncbi.nlm.nih.gov/pmc/articles/PMC5952231/>
128. Karban R, Agrawal AA. Herbivore Offense. Annual Review of Ecology and Systematics. 2002;33(1):641-64.
129. Thompson J. What we know and do not know about coevolution: insect herbivores and plants as a test case. Herbivores: between plants and predators. 1999;7-30.
130. Agrawal AA. Overcompensation of plants in response to herbivory and the by-product benefits of mutualism. Trends in plant science. 2000;5(7):309-13.
131. Garcia LC, Eubanks MD. Overcompensation for insect herbivory: a review and meta-analysis of the evidence. Ecology. 16 déc 2018;ecy.2585.
132. Järemo J, Tuomi J, Nilsson P, Lennartsson T. Plant Adaptations to Herbivory: Mutualistic versus Antagonistic Coevolution. Oikos. févr 1999;84(2):313.
133. Adroit B, Wappler T, Terral J-F, Ali AA, Girard V. Bernasso, a paleoforest from the early Pleistocene: New input from plant–insect interactions (Hérault, France). Palaeogeography, Palaeoclimatology, Palaeoecology. mars 2016;446:78-84.
134. Bouchal JM, Güner TH, Denk T. Middle Miocene climate of southwestern Anatolia from multiple botanical proxies. Climate of the Past Discussions. 9 juill 2018;1-30.
135. Cohen KM, Finney SC, Gibbard PL, Fan J-X. The ICS International Chronostratigraphic Chart. 2013;(36):199-204.

- 1
2
3
4
5
6
7
8
9
10
11
12
13
14
15
16
17
18
19
20
21
22
23
24
25
26
27
28
29
30
31
32
33
34
35
36
37
38
39
40
41
42
43
44
45
46
47
48
49
50
51
52
53
54
55
56
57
58
59
60
136. DeChaine EG, Martin AP. Using coalescent simulations to test the impact of quaternary climate cycles on divergence in an alpine plant-insect association. *Evolution*. 2006;60(5):1004.
137. Milne RI. Northern Hemisphere Plant Disjunctions: A Window on Tertiary Land Bridges and Climate Change? *Annals of Botany*. 21 juin 2006;98(3):465-72.
138. Suc J, Popescu S. Pollen records and climatic cycles in the North Mediterranean region since 2.7 Ma. *Special publication-geological Society of London*. 2005;247:147.
139. Biltekin D, Popescu S-M, Suc J-P, Quézel P, Jiménez-Moreno G, Yavuz N, et al. Anatolia: A long-time plant refuge area documented by pollen records over the last 23million years. *Review of Palaeobotany and Palynology*. avr 2015;215:1-22.
140. Bińka K, Nitychoruk J, Dzierzek J. *Parrotia persica* C.A.M. (Persian witch hazel, Persian ironwood) in the Mazovian (Holsteinian) Interglacial of Poland. *Grana*. déc 2003;42(4):227-33.
141. Jiménez-Moreno G, Fauquette S, Suc J-P. Miocene to Pliocene vegetation reconstruction and climate estimates in the Iberian Peninsula from pollen data. *Review of Palaeobotany and Palynology*. oct 2010;162(3):403-15.
142. Theissinger K, Bálint M, Feldheim KA, Haase P, Johannesen J, Laube I, et al. Glacial survival and post-glacial recolonization of an arctic-alpine freshwater insect (*Arcynopteryx dichroa*, Plecoptera, Perlodidae) in Europe. *Journal of Biogeography*. 2013;40(2):236-48.
143. Li C, Wang W, Sun Q, Li F, Zhang J, Wang X, et al. Stratigraphical sequence of diato- maceous beds within Shanwang Formation, Linqu Country, Shandong Province, China. *植物学报*. 20 déc 2000;17:247-51.
144. Meischner D. Europäische Fossilagerstätten. In: *Der pliozäne Teich von Willershausen am Harz*. Pinna G. New-York, USA: Dieter Meischner; 2000. p. 223-8 and 261.
145. Vinken R. Kurzer Überblick über die Geologie der Umgebung von Willershausen. *Bericht der Naturhistorischen Gesellschaft zu Hannover*. 1967;115:5-14.
146. Aide TM. Dry Season Leaf Production: An Escape from Herbivory. *Biotropica*. 1992;24(4):532-7.
147. Alarcón R, Waser NM, Ollerton J. Year-to-year variation in the topology of a plant-pollinator interaction network. *Oikos*. 2008;117(12):1796-807.
148. Bale JS, Masters GJ, Hodkinson ID, Awmack C, Bezemer TM, Brown VK, et al. Herbivory in global climate change research: direct effects of rising temperature on insect herbivores. *Global Change Biology*. 2002;8(1):1-16.
149. Thompson JN, Fernandez CC. Temporal Dynamics of Antagonism and Mutualism in a Geographically Variable Plant-Insect Interaction. *Ecology*. 2006;87(1):103-12.

Figures

Figure 1

Figure 2

1
2
3
4
5
6
7
8
9
10
11
12
13
14
15
16
17
18
19
20
21
22
23
24
25
26
27
28
29
30
31
32
33
34
35
36
37
38
39
40
41
42
43
44
45
46
47
48
49
50
51
52
53
54
55
56
57
58
59
60

Plant species	Type	Locations	DT297 measurements			Wilcoxon test** (p.value)		
			Surface area	Length	Width*	Hyrcanian	Shanwang	Yixing
P. persica	Fossil	Willershausen	4.54	5.85	0.68	0.27	0.26	0.59
	Present	Hyrcanian	3.47	4.89	0.56			
P. subaequalis	Fossil	Shanwang	3.61	4.62	0.59	0.82	0.53	0.35
	Present	Yixing	4.75	6.33	0.67			
Standard deviation ($\alpha=5\%$)								
			0.89	0.71	0.09			
			1.87	2.53	0.07			
			0.87	0.79	0.09			
			2.01	1.35	0.08			

Figure 3

Figure 4: Artistic representation of both *Parrotia persica* (left) and *Parrotia subaequalis* (right) scarred by the damage type D1297. Illustration has been produced by Mr. Dinghua Yang from Nanjing Institute of Geology and Palaeontology, Nanjing, China.

Figure 4

1
2
3
4
5
6
7
8
9
10
11
12
13
14
15
16
17
18
19
20
21
22
23
24
25
26
27
28
29
30
31
32
33
34
35
36
37
38
39
40
41
42
43
44
45
46
47
48
49
50
51
52
53
54
55
56
57
58
59
60

Figures captions

1
2 Figure 1: Eurasian locations including both fossil and modern occurrences of ~~the genera~~ *Parrotia* ^{that contain examples} which imply the presence
3 of the damage type DT297.

4
5 The distribution for *P. subaequalis* in China has been drawn in one large area for a better visibility. For real, the distribution
6 of *P. subaequalis* in ^{the} Yangtze River valley is very fragmented, only small ^{isolated} separate populations occur in the valley. More
7 details ^{are provided by} Geng et al., 2015; Li & Zhang 2015.

8
9
10
11 Figure 2: Damage type DT297 on every type of leaves ^{is attributed} belong to *Parrotia* ~~genera~~.

12
13 A1-A6 Fossil specimens of *P. pristina* from the Pliocene of Willershausen ~~outcrop~~, Germany. The material is deposited in
14 the Geoscience Centre, University of Göttingen (GZG), Germany. B1-B2 Modern specimens of *P. persica* from the
15 Hyrcanian forest (northern Iran), more details ^{are provided by} in Adroit et al. (2018b). C1-C2 Modern leaves of *P. subaequalis* from the
16 Yixing forest, Yangtze River area, eastern China. D1-D4 Fossil specimens of *S. subaequalis* (= ^{synonym} analogue of *P. subaequalis*)
17 from the mid-Miocene of Shanwang ~~outcrop~~, China. The material is deposited in the collection belong ^{ing} to Nanjing Institute
18 of Geology and Paleontology (NIGPAS), China. Black bars represent 2.5mm.

19
20
21
22 Figure 3: Box-plots based on the comparison of the average area of DT297 per locality.

23
24 Green boxes ^{represent} mean modern leaves, brown boxes ^{denote} mean fossil leaves. At the top there is a table with the whole
25 measurements made on DT297. ^{each} Wilcoxon test comparison has been made on measurements and the results ^{concerning} about surface
26 area comparison between every location are presented to the right side. Surface area has been chosen as it ^{is} directly
27 include the length and the width. There is no significant difference between every sites ^{on} comparison ($\alpha=1\%$).
28
29
30

31 Figure 4: Artistic representation of both *Parrotia persica* (left) and *Parrotia subaequalis* (right) ^{bearing} with the damage type DT297.

32
33 ^{Their} illustration has been ^{was} produced by Mr. Dinghua Yang from ^{the} Nanjing Institution ^e of Geology and Paleontology, Nanjing,
34 China.

World Mercator projection

Appendix C

Dear Prof Professor Elizabeth Harper,

Thank you for let us resubmit our manuscript in Royal Society Open Science.

We carefully read all the comments and paid attention to completely review every single details mentioned by the reviewer 1 as you suggested to do it. We also answer to each specific comment in that sense, see below. We obtain an answer from Geowissenschaftliches Museum Göttingen and had the confirmation that the holotype for DT297 is still the same one, originally chose by Straus (1977). We completely changed and corrected some minor mistakes into the references.

Moreover, as you mentioned the main notes from Dr Steven McLoughlin, I contacted him in order to make better improvement to the manuscript. He was very kind to me and take his time to help me in the revision, according to his comments.

We strongly believe that our manuscript fits with your requirements and the one of the journal.

NB: I believe the reviewer 2 did not review the last version of our manuscript as he/she makes references to lines number from the former manuscript version (which are not existing in the new one that we submitted to you). Also some comments has already been corrected in the previous version.

Please let me know if you have any further questions or requirements for this manuscript.

Reviewer 1:

Note: Lines numbers refer to the ones from the PDF document of the reviewer.

Comment: "The last sentence of the 1st paragraph of the introduction does not hang together very well with the preceding text. It needs some connection between the sentences dealing with ecosystem balance / quantification of herbivory damage, and the last sentence dealing with historical establishment of a catalogue of damage types."

Answer: We absolutely understood this comment, however this last sentence sounded quite repetitive with a similar one in the M&M. Then we deleted the last sentence of the 1st paragraph as mentioned by the reviewer.

Comment: "Throughout the text, the distinctive damage features on *Parrotia* species are described as "new". This is misleading, as the damage was described in 1977 by Straus under a formal ichnospecies name. The only thing that is "new" about the illustrated damage is that it has a new designation "DT297" in a proposed future revision of the Labandeira et al. (2007) damage catalogue."

Answer: We originally meant that this damage was new for the topic of the plant-insect interactions. This topic became more important in paleontology and now can be use homogenously throughout all fossil locations in the world since the publication of the "Guide to Insect (and Other) Damage Types on Compressed Plant Fossils" (Labandeira et al., 2007). This guide is the main reference for plant-insect interaction in the fossil records and never referenced the damage we observed, that is one of the reasons we called it "new". Moreover, this damage has been observed in modern *Parrotia* (while Straus 1977 said that this damage does not exist anymore on the modern *Parrotia*) and on fossil *Parrotia* in China, which was even unknow at the life period of Dr Straus; that is another reason why we called it "new".

However, we understand the suggestion of the reviewer and we recognize that it could be misleading for the reader. Consequently we only specified that the damage was new for the guide from Labandeira et al. (2007) (section 5.1) and we deleted all of the “new damage type” from the manuscript to avoid wrong interpretation and to be fair with the first observation from Straus (1977).

Comment:” Throughout the text, the fossil *Parrotia* leaves are described as “analogues” of modern *Parrotia* plants. I think this is the wrong choice of words. They might have occupied analogous environments, but the plants themselves might be better described as “relatives”. ».

Answer: According to the literature cited in the manuscript, the Hyrcanian forest in northern Iran is defined as a modern analogue forest of the Eurasian paleoforests. However, it is true that it is not clearly sure that the modern plant species themselves, such as *Parrotia persica*, are strictly similar in terms of ecology to their ‘ancestor’ such as *Parrotia pristina*. Consequently, instead of referring to these in terms of “analogues”, it is more appropriate to refer to these as “relative” as the reviewer suggested, reflecting the affinity between *Parrotia* species from the fossil and the modern records.

Comment: “In section 3.1, the authors present a narrative of the “migration” of *Parrotia* across Eurasia-North America in the Cenozoic. Yet all the fossil record can really tell us is that a taxon is present at a particular place at a particular time. Absence of fossils does not necessarily mean absence of the taxon in the region – just that it hasn’t been preserved at that specific site. Moreover, many regions lack any fossil assemblages of the relevant age. In general, a narrative has been created based on a range of assumptions about how the plants responded to climate change, mountain belt uplift, etc. One can never really confirm the direction of “migration”/range expansion unless one has fossils from every year of the Cenozoic preserved in every region of the Northern Hemisphere. »

Page 7 - L28-30: “*Parrotia* was present in East Asia and possibly North America during the Eocene and migrated to western Eurasia across Central Asia during the Oligocene (32). During the early Oligocene the genus was present in Kazakhstan from where it disappeared during the Miocene (33).”

Answer: *Parrotia* was present in East Asia and possibly North America during the Eocene and most likely spread to western Eurasia across Central Asia during the Oligocene. This biogeographic patterns has been reported many times in the palaeobotanical literature (see e.g. Walther, 1994; Mai, 1995; Corbett & Manchester, 2004) and has been confirmed using molecular data (see, for example, Renner et al., 2016; Jiang et al., 2019, among many others).

However, to make it easier for the readers, we changed the word “migration” by “spread”. We believe this new terminology will avoid any plausible misunderstanding from the readers.

Walther H 1994 Invasion of Arcto-Tertiary elements in the Paleogene of central Europe. Pages 239–250 in MC Boulter, HV Fisher, eds. Cenozoic plants and climates of the Arctic. Vol 27. NATO ASI series. Springer, Berlin.

Mai, D.H., 1995. Tertiäre Vegetationsgeschichte Europas. Fischer, Jena.

Corbett, S.L., Manchester, S.R., 2004. Phytogeography and fossil history of *Ailanthus* (Simaroubaceae). *Int. J. Plant Sci.* 165, 671–690.

Renner, S.S., Grimm, G.W., Kapli, P., Denk, T., 2016. Species relationships and divergence times in beeches: new insights from the inclusion of 53 young and old fossils in a birth–death clock model. *Philos. Trans. R. Soc. B Biol. Sci.* 371, 20150135. <https://doi.org/10.1098/rstb.2015.0135>

Jiang, X.-L.; Hipp, A.L.; Deng, M.; Su, T.; Zhou, Z.-K.; Yan, M.-X. East Asian origins of European holly oaks (*Quercus* section *Ilex* Loudon) via the Tibet-Himalaya. *J. Biogeogr.* 2019.

Comment: “In section 4.1, the damage on *Parrotia* is described as a “mine”, yet it is also described as consisting of several “holes” or “chain of holes” (suggesting hole feeding, and in section 5.1 it is described as a form of “skeletonization”. There needs to be some consistency in the description and categorization of this damage type.”

Answer: It is true that DT297 is, according to the guide (Labandeira et al., 2007) classified as a skeletonization. Originally the terms “holes, chain of hole” was used to describe the shape of the damage; however, “HOLE” is also the name of one functional feeding group in the guide (Labandeira et al., 2007). Consequently, we understand that it could be confusing for the reader, then we erased all those terms and only mentioned “skeletonization” through the text.

Comment: “I would hope that, to maintain some consistency between formal ichnotaxonomic schemes and the informal damage categories employed by Labandeira et al. 2007, the holotype chosen for *Phagophytichnus catellarius* is the same specimen chosen as the reference specimen for DT297. If not, there may be issues in the future about how these damage features are named. »

Answer: We finally obtained the information that we required after read this comment. Dr Alexander Gehler and Mrs Lina Leschner (Geowissenschaftliches Museum, Göttingen) confirmed that the original fossil holotype chose for *Phagophytichnus catellarius* is the same that the one on Supplementary 1. Then, we adapted our text in the section 4.3 The specimen of DT297, also we used a new picture of this fossil for the supplementary file 1.

Comment: “The authors need to be careful with their use of Ma and Myrs. Ma is used to denote a point in time (20 Ma = 20 million years ago), whereas Myrs describes a span of time (20 Myrs = a 20-million-year interval).”

Answer: We corrected all those abbreviations through the manuscript following the rule wrote by the reviewer.

Page 7 - L45-46: “Nowaday"s, *P. persica* exists only in the Hyrcanian forest south of the Caspian Sea (Iran, Azerbaijan).”

We kept this information in the section 3.1 and deleted from the introduction, it was repetition of information as suggested by the reviewer.

Page 10 - L25: As we wrote previously, we changed the photography in the supplementary data and we confirm the modern reference of the holotype is the same that old one from Straus (1977).

Page 10 - L47/49: We adapted all the nomenclature through the text to avoid any misunderstanding.

Page 11 – L24: We modified the two sentences and simplified into one to avoid any repetitions of information.

Page 11 – L38 : We modified the sentence in order to emphasize the concordance of the lineage between the *Parrotia* species.

References: Apologizes for the reference formatting. We corrected all the references by using the official plugin from the Royal Society Open Science.

Reviewer 2:

Comment: English of the text is very weak. There are too many grammatical errata. I strongly suggest that a native speaker needs to read the text before resubmission.

Answer: We reviewed all the English weakness with the full review correction from the reviewer 1.

Comment: What is sampling frame mentioned in line 78? You should explain more how you did your sampling. Did you collect leave on a random-based framework? How many sites/locations were assessed during your sampling? How many leaves of other species have been included in your sampling? This phenomenon is very rare in the modern leaves (as you already insisted in your MS), then one could hypothesize that the same herbivory could also probably happen on other trees like Zelkova, Alnus, Fagus or Carpinus if enormous number of leaves of the latter species would be collected in the sampling plots. Please

Answer: All these information can easily be found in Adroit et al. 2018b; as this is not the point of the present manuscript, we did not detail the overall protocol from another study. We slightly modified the sentence to be more clear.

Comment: I am still not very confident that herbivory trace on fossil and modern leaves are really coming from same type of herbivore. Could you discuss more about other similar type of traces?

Answer: We have taken great pains to state carefully what the data do and do not show, and we have made changes to the manuscript to prevent misinterpretations. The core result from our paper is that the observed damage on the fossil plants (*Parrotia*) compared to the modern, based on suitable statistical analysis.

We added a sentence in the conclusion to make it more clear.

Comment: I cannot see any reason to include lines 218 to 222 in the discussion part. This is part of your result.

Answer: We modified this sentence according to the comment from the reviewer 1.

Appendix D

Dear Prof Elizabeth Harper,

Thank you to let us the chance to resubmitted this paper during the last months. The reviews from both reviewer has been excellent to improve my point of view of our results. Despite my young age and lower experience into manuscript submission processes than my colleagues, I must to admit that the experience with Royal Society Open Science, supervised by Prof Andrew Dunn and yourself has been very pleasant to me. It is definitely my best “first-author experience” that I never had. I also underline this point because as you probably saw in my emails exchange with Andrew Dunn, I have an internal conflict of author to handle, as the Prof Bo Wang from Nanjing, who is supposed to take the fees in charge (that is why he is last author), changed his mind at the last moment. In parallel, with the nice advices from Prof Dunn, I join the “Discretionary Waiver Application” form to my submission and crossed my fingers to make my situation better than right now. According to your comment which I completely agree, it is obvious that I wrote the name of the Dr McLoughlin in the acknowledgments, I also tell him thank you very much personally.

Again, I use the opportunity of this letter to thank all of the RSOS team and especially both of the Associate Editors.

Benjamin Adroit, PhD